# LEXICAL DIVERSITY-AWARE RELEVANCE ASSESSMENT FOR RETRIEVAL-AUGMENTED GENERATION

## ABSTRACT

Despite their extensive applications, large language models trained on vast historical datasets still struggle with hallucination issues, particularly when addressing open-ended, factual, and commonsense questions. In contrast, Retrieval-Augmented Generation (RAG) methods have proven effective in enhancing large language models' responses to such inquiries, making them a focal point of research. However, previous RAG approaches overlook the lexical diversity of queries, hindering their ability to achieve a granular relevance assessment between queries and retrieved documents, resulting in suboptimal performance. In this paper, we introduce a Lexical Diversity-aware RAG (DRAG) model, comprising a Diversity-sensitive Relevance Analyzer (DRA) and a Contrastive Relevance Calibration Module (CRC). Specifically, DRA decouples and assesses the relevance of different query components (words, phrases) based on their levels of lexical diversity, ensuring precise and comprehensive document retrieval. According to the DRA assessment, CRC further emphasizes the pertinent knowledge of the retrieved relevant documents through contrastively eliminating the adverse effects of irrelevant contents. By integrating DRA and CRC, the proposed method effectively retrieves relevant documents and leverages their pertinent knowledge to refine the original results and generate meaningful outcomes. Extensive experiments on widely used benchmarks demonstrate the efficacy of our approach, yielding a 12.4% accuracy improvement on HotpotQA.

## 1 INTRODUCTION

The rapid development of large language models (LLMs) has led to widespread deployment across various fields, including conversational assistants (Achiam et al., 2023; Touvron et al., 2023; Dubey et al., 2024), medical diagnosis (Yang et al., 2024b; Li et al., 2024), and code generation (Wei et al., 2023; Luo et al., 2023). However, LLMs rely solely on their training parametric knowledge for inference, which frequently results in issues such as factual hallucinations, outdated information, and low interpretability (Mallen et al., 2022; Huang et al., 2023; Ji et al., 2023), particularly in tasks requiring open-domain knowledge or real-time information (Shuster et al., 2021; Li et al., 2023).

Retrieval-Augmented Generation (RAG) (Lewis et al., 2020) has demonstrated significant effectiveness in improving factual accuracy by integrating external retrieval knowledge to enhance LLM generation. Typically, RAG methods first utilize a retriever to acquire potentially relevant documents based on an input query in the retrieval stage, and subsequently extract key information from these documents to augment LLMs in the generation stage. Self-RAG (Asai et al., 2023) trains LLM to assess whether the retrieved documents are related to the input query to improve the retrieval validity. RA-DIT (Lin et al., 2023) fine-tunes both the LLM and the retriever, concurrently enhancing their retrieval and generation performance. Methods such as RECOMP (Xu et al., 2024) and SuRe (Kim et al., 2024) summarize retrieved documents to extract key content to mitigate the impact of irrelevant text on the model during the generation stage.

However, previous RAG methods struggle to establish a granular relevance assessment between queries and retrieval documents, leading to an under-utilization of external relevant knowledge. As shown in Figure 1, in the **retrieval stage**, existing RAG approaches assess the relevance of retrieved

Figure 1: The challenge of previous methods. In retrieval, the lexical diversity results in differing retrieval complexities. In generation, reductive summarization induces information omission.

documents based on single criteria, neglecting the lexical diversity of different fine-grained query components (words or phrases): **(1)** Some components, such as proper names, consistently remain in a fixed form and can be assessed for relevance straightforwardly. **(2)** Some components may be expressed in various lexical forms, such as "occupation" being expressed as "profession", a specific job like "actress", or even as achievements like "Academy Award", complicating the relevance assessment. **(3)** Beyond the original query components, supplementary information like "American celebrities" in relation to 'Hattie McDaniel's occupation' may aid in relevance assessment and also exhibit a certain degree of lexical diversity. As a result, this variation across different query components causes documents with partially similar phrases to be incorrectly considered highly relevant, while documents containing strongly related content but expressed in different ways are overlooked. In the **generation stage**, owing to the prevalent entanglement of irrelevant text with relevant information, existing RAG methods relying on reductive summarization inaccurately extract fine-grained relevant information, while approaches that require further training are resource-intensive and heavily data-dependent. Therefore, a sound granular relevance assessment mechanism is essential at both the retrieval stage and generation stage, facilitating the acquisition of relevant knowledge and further improving LLM performance.

In this paper, we propose a Lexical Diversity-aware RAG (DRAG) deployed with a Diversity-sensitive Relevance Analyzer (DRA) and a Contrastive Relevance Calibration Module (CRC), effectively harnessing the relevant external knowledge. To handle the lexical diversity, DRA introduces distinct relevance assess criteria for different query components, enhancing granular query-document matching. We investigate the varied lexical diversity attributes and assessment mechanism, and prompt the analyzer to achieve query decomposition and relevance evaluation. As the degree of lexical diversity increases, the evaluation criteria become more flexible and detailed, ensuring the accuracy and adequacy of the returned documents. To strengthen the granular relevant knowledge, CRC minimizes the impact of irrelevant information contained in the retrieved documents through decoding adjustment. It first approximates the irrelevant information through constructing a noisy reference document based on the relevant assessment of DRA. The noisy reference document that strongly interferes with the generation process, is then fed into the generation model, where the original generation result is further refined by eliminating the noisy reference decoding distribution. This contrastive calibration strategy could effectively avoid information omission from unsuitable reduction and the resource consumption associated with retraining. Thus, the proposed method could retrieve comprehensive relevant documents through the granular assessment criteria and promote relevant knowledge aggregation for LLM generation.

To evaluate the effectiveness of our proposed method, we perform extensive experiments on commonly used open-domain question answering benchmarks. The experimental results demonstrate that our approach efficiently extracts retrieval content that is semantically aligned with the query, leading to significant improvements in the factual accuracy of RAG. Specifically, on HotpotQA, our method performs a 12.4% accuracy boost over the second-best method.

## 2 RELATED WORK

The Retrieval-Augmented Generation (RAG) framework, introduced by (Lewis et al., 2020), incorporates relevant information from external document repositories to improve the performance

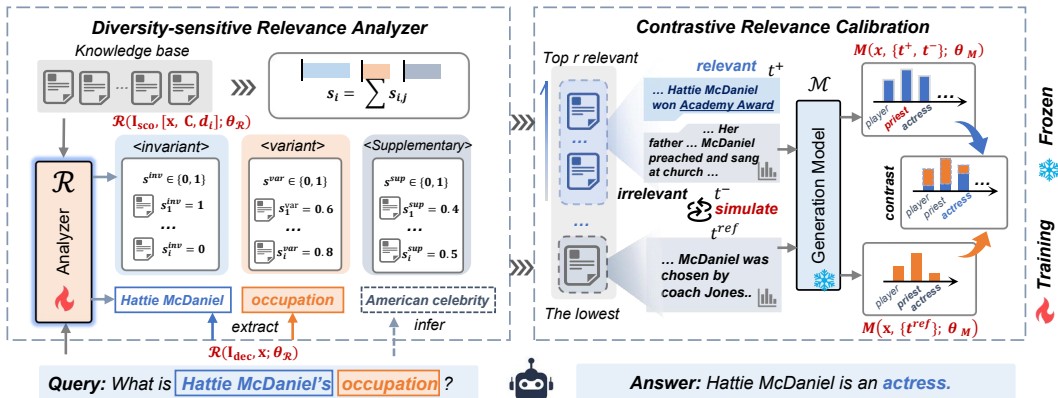

Figure 2: An overview of our DRAG. DRA first decouples different query components based on their lexical diversity and conducts a precise relevance analysis of retrieved documents. CRC then calibrates the model's decoding process by contrasting it with outputs under noise information.

of pre-trained large language models (LLMs) during generation. This approach has demonstrated significant efficacy in knowledge-intensive tasks. Research on RAG has primarily focused on enhancing the model's ability to retrieve and utilize external knowledge through training (Guu et al., 2020; Izacard et al., 2023; Borgeaud et al., 2022; Wei et al., 2024). For instance, Self-RAG (Asai et al., 2023) focuses on improving retrieval accuracy by training LLMs to evaluate the relevance of retrieved documents in relation to the input query. RA-DIT (Lin et al., 2023) fine-tunes both the LLM and the retriever, simultaneously improving their performance in retrieval and generation tasks. Other methods, such as RECOMP (Xu et al., 2024) and SuRe (Kim et al., 2024), aim to summarize retrieved documents to mitigate the impact of irrelevant content during generation. However, these approaches commonly depend on coarse relevance assessments, failing to account for the lexical diversity of granular query components, which can hinder their effectiveness in fine-grained retrieval and generation. Moreover, imprecise reductive compression often results in the loss of key information. In contrast, our method introduces granular relevance assessment criteria that address the lexical diversity of query components, and employs contrastive calibration to mitigate the influence of irrelevant information. This approach promotes comprehensive retrieval and efficient utilization of relevant knowledge, without the need for extensive training or reductive summarization.

## 3 METHODOLOGY

We propose a Lexical Diversity-aware RAG (DRAG) framework, which incorporates a fine-grained relevance evaluation mechanism across both the retrieval and generation stages to sufficiently leverage the relevant knowledge. DRAG consists of a Diversity-sensitive Relevance Analyzer (DRA) and a Contrastive Relevance Calibration Module (CRC), designed to operate with minimal training resources. In this section, we first present the problem formalization and an overview of DRAG in 3.1, followed by a detailed introduction to the DRA module in 3.2 and the CRC module in 3.3.

### 3.1 PROBLEM FORMALIZATION AND OVERVIEW

We conform to the standard setup of RAG (Lewis et al., 2020; Asai et al., 2023) and the Open-Domain Question Answering task. In the retrieval stage, given an input query $\mathbf{x}$, the retriever returns the top $k$ retrieved documents $\mathbf{D} = \{d_1, d_2, \ldots, d_k\}$ based on embedding similarity. Advanced RAG methods further assess or process the documents in $\mathbf{D}$ based on their relevance and construct relevant documents set $\mathbf{D}^{\text{rel}}$ feeding into the generation model $\mathcal{M}$. In the generation stage, the model $\mathcal{M}$ is tasked to predict an answer $\mathbf{y}$ to the given query $\mathbf{x}$ based on $\mathbf{D}^{\text{rel}}$ and its parametric knowledge.

**Overview.** The overview process of our DRAG is illustrated in Algorithm 1. In the retrieval stage, The DRA module takes the query $\mathbf{x}$ and retrieved documents $\mathbf{D} = \{d_1, d_2, \ldots, d_k\}$ as input, to further extract more related documents. It first decomposes the query into multiple components $\mathbf{C}$ and evaluates the relevance $\mathbf{s}_{i,j}$ between the documents $d_i$ and $j$-th query component. After the

---

**Algorithm 1** DRAG: Lexical Diversity-Aware Retrieval-Augmented Generation

---

**Require:** Query $x$, Documents $\mathbf{D} = \{d_1, d_2, \ldots, d_k\}$, DRA Module $\mathcal{R}$, Generation Model $\mathcal{M}$;
    Random text $t_{\text{random}}$, Fixed noise text $t_{fix}$
    /* The Retrieval Stage */
 1: **for** each $d_i \in \mathbf{D}$ **do**
 2:     Generate query components $\mathbf{c} \leftarrow \mathcal{R}(\mathrm{I}_{\text{dec}}, \mathbf{x}; \theta_{\mathcal{R}})$            ▷ Query Decoupling (§ 3.2)
 3:     Generate relevance analysis $\mathbf{s} \leftarrow \mathcal{R}(\mathrm{I}_{\text{sco}}, [x, \mathbf{C}, d_i]; \theta_{\mathcal{R}})$    ▷ Relevance Assessment (§ 3.2)
 4:     Calculate final score $s_i \leftarrow \sum \mathbf{s}$
 5: **end for**
 6: Sort documents $\mathbf{D}^{'} = \{d_1^{'}, d_2^{'}, \ldots, d_k^{'}\} \leftarrow \text{Sort}(\mathbf{D}; s^{\text{rel}})$ according to final score $s^{\text{rel}}$
 7: Select top $r$ documents $\mathbf{D}^{\text{rel}} = \{d_1^{'}, d_2^{'}, \ldots, d_r^{'}\}$
    /* The Generation Stage */
 8: Construct noise text $t_{ref} = d_k^{'}$                ▷ Contrastive Relevance Calibration (§ 3.3)
 9: Adjust decoding process $y \leftarrow \mathcal{M}(x, d_i; \theta_{\mathcal{M}}) - \gamma * \mathcal{M}(x, t_{\text{ref}}; \theta_{\mathcal{M}})$
10: **return** $y$

---

granular relevance assessment, the overall relevance summarizes the relevance of all components and selects the top $r$ highest-ranked documents to achieve further LLM enhancement. Based on the granular relevance score of ADR, the Contrastive Relevance Calibration Module (CRC) then adjusts the decoding distribution of the generation model $\mathcal{M}$, to exclude the negative impacts of extraneous information and generate reasonable results.

## 3.2 DIVERSITY-SENSITIVE RELEVANCE ANALYZER

Existing approaches, which apply a single relevance criterion, fail to capture the granular relevance between queries and retrieved documents, disregarding the effects of lexical diversity. Therefore, we propose the Diversity-sensitive Relevance Analyzer (DRA), which decouples the relevance assessment process to accommodate varying degrees of lexical diversity among query components. DRA decomposes the query into distinct components and diversely evaluates the intrinsic relevance between each component and the retrieved documents, employing tailored criteria according to the extent of diversity. This enables a more accurate and fine-grained relevance assessment between the retrieved documents and the full query.

**Lexical Diversity-Driven Query Decoupling.** To perform granular relevance assessment, We first explore the different attributes of lexical diversity and the corresponding assessment mechanisms, guiding the analyzer to perform query decomposition. Specifically, we categorize the query components into three attributes: $\mathrm{A} = \{< Invariant >, < Variant >, < Supplementary >\}$, taking the query "*What is Portland the capital of?*" as an example:

- $< Invariant >$: Components without lexical diversity that are directly extracted from the query. The invariant component of the example query is "*Portland*", whose expression cannot be altered.

- $< Variant >$: Components with lexical diversity that are directly extracted from the query. The variant component of the example query is "*Capital*", which could be expressed with synonyms such as "*Administrative center*" or "*Seat of government*".

- $< Supplementary >$: Components not explicitly mentioned in the query but can be reasonably inferred to supplement and enhance relevance assessment, which is not mandatory and demonstrates a significant degree of lexical diversity. A possible supplementary component of the example query is "*State or country*", which can be inferred from the query and is beneficial for relevance assessment.

Based on the attribute categorization, we train the DRA module $\mathcal{R}$ to decouple various query components $\mathbf{c} = \{< c_1, a_1 >, < c_2, a_2 >, \ldots, < c_n, a_n >\}$ and assign the attribute $a_j \in \mathrm{A}$ to each component $c_j$. $n$ represents the total number of elements, which is determined by $\mathcal{R}$:

$$\mathbf{c} = \mathcal{R}(\mathrm{I}_{\text{dec}}, \mathbf{x}; \theta_{\mathcal{R}}) \tag{1}$$

where $I_{dec}$ is the instruction for $\mathcal{R}$ to generate decoupling components (*"You are an assistant in extracting key components from a given question."*), and $\theta_{\mathcal{R}}$ is the fine-tuned parameters of $\mathcal{R}$. By decoupling the query and assigning distinct attributes to its components, we can effectively account for the lexical diversity across different elements, enabling a tailored relevance assessment.

**Granular Relevance Assessment.** To accurately assess the relevance between each component and the retrieved documents, we further apply granular assessment criteria tailored to the attributes of different components. As the level of lexical diversity increases, the evaluation criteria are progressively refined and made more stringent, ensuring both the precision and comprehensiveness of the retrieved documents. Specifically, we prompt the DRA module with instruction $I_{sco}$ (*"You are an assistant in scoring documents based on a given question and its components."*) to granular assess the relevance between the retrieval document $d$ and each component $c_j$:

$$\mathbf{s}_i = \mathcal{R}(I_{sco}, [\mathbf{x}, \mathbf{C}, d_i]; \theta_{\mathcal{R}}) \tag{2}$$

where $\mathbf{s}_i = \{<s_{i,1}, e_{i,1}>, <s_{i,2}, e_{i,2}>, \ldots, <s_{i,n}, e_{i,n}>\}$, $s_{i,j}$ is the relevance score of the $j$-th query components associated with the document $d_i$, and $e_j$ denotes the corresponding explanation.

For invariant component $c_j$ whose attribute $a_j =< Invariant >$, $\mathcal{R}$ applies strict evaluation criteria $\sigma_1$ and assigns a binary relevance score. If the retrieved document $d$ explicitly mentions $c_j$, the score $s_{i,j}$ is set to 1; otherwise, it is 0. For variant and supplementary components $c_j$ with attributes $a_j =< variant >$ or $< supplementary >$, the DRA module $\mathcal{R}$ is fine-tuned to apply more flexible criteria $\sigma_2$, and provides a continuous score $s_{i,j} \in [0, 1]$. The specific implementation of $\sigma_1, \sigma_2$ is detailedly discussed in the Appendix Section B. We compute the sum of all query components $\mathbf{s}_i = \sum_{j=1}^{n} s_{i,j}$, indicating the overall relevance degree between the query $\mathbf{x}$ and the $i$-th document $d_i$.

Finally, the top $r$ retrieved documents with the highest overall relevance score $\mathbf{s}_i$ are then sent to the model $\mathcal{M}$ to enhance the generation.

The DRA module accounts for lexical diversity across query components, facilitating a fine-grained and accurate relevance assessment between the query and retrieved documents. This ensures the retrieval of documents that are highly relevant to the query. The module is fine-tuned using data tailored for both query decomposition and granular relevance evaluation.

### 3.3 CONTRASTIVE RELEVANCE CALIBRATION

Existing methods attempt to eliminate irrelevant text from retrieved documents through reductive summarization, but they typically lead to the omission of critical information due to imprecise reduction. Therefore, we propose a Contrastive Relevance Calibration (CRC). Inspired by the inference-time augmentation method contrastive decoding (Li et al., 2022), We calibrate the model's decoding distribution by comparing it against a noise reference, rather than performing reduction on the input. This approach effectively minimizes the influence of granular noise while avoiding the omission of important information.

**Motivation.** Usually the retrieved document $d_i$ contains the relevant knowledge $t^+$ and a significant amount of irrelevant noise text $t^-$. Previous reduction-based RAG methods normally lower the impact of the irrelevant noise and generate the final results through the reductive compression *reduction*(.) as follows:

$$\mathbf{y} = \mathcal{M}(\mathbf{x}, reduction(\{t^+, t^-\}); \theta_{\mathcal{M}}) \tag{3}$$

$$\mathcal{M} = \arg\min D_{KL}\left(\mathcal{M}(\mathbf{x}, reduction(\{t^+, t^-\}); \theta_{\mathcal{M}}) \| \mathcal{M}(\mathbf{x}, t^+; \theta_{\mathcal{M}})\right) \tag{4}$$

However, reductive compression often leads to the omission of critical information when granular relevance assessment is lacking, adversely affecting the accuracy of the output. Additionally, training-based generation methods aim to introduce more relevant data for updating the parameter $\theta_{\mathcal{M}}$, which can result in excessive resource consumption. Therefore, we come up with an efficient generation method inspired by constrastive decoding (Li et al., 2022).

**Contrastive Calibration.** Unlike other methods that perform reductive processing on the input document $d$, we calibrate the model's decoding process by contrasting it with output under irrelevant noisy text, without resorting to reductive operations and risking information omission from the retrieval documents. Specifically, we aim to mitigate the impact of irrelevant noise in the model's output distribution:

$$\mathbf{y} = \mathcal{M}(x, \{t^+, t^-\}; \theta_{\mathcal{M}}) - \mathcal{M}(x, t^-; \theta_{\mathcal{M}}) \tag{5}$$

Since accurately capturing irrelevant text is challenging, we instead construct a reference noise text $t^{\text{ref}}$ to simulate the irrelevant text $t^-$ during the generation process. Based on the DRA's assessment results, we select the document with the lowest relevance score as the noisy reference document. This document represents irrelevant information that may contain similar vocabulary but differs semantically from the query, thereby reinforcing the relevant information and eliminating extraneous content. The equation 5 can be expressed as follows:

$$\mathbf{y} = \mathcal{M}(x, \{t^+, t^-\}; \theta_{\mathcal{M}}) - \gamma \mathcal{M}(x, t^{ref}; \theta_{\mathcal{M}}), \quad t^{ref} = \arg\min_{d_i \in \mathbf{D}} s_i \tag{6}$$

where $\gamma$ denotes the calibration weight.

Through granular relevance assessment based on lexical diversity, the ADR can retrieve more relevant documents, while the CRC further emphasizes pertinent information within these documents to generate meaningful results. By integrating the ADR and CRC, the proposed DRAG approach effectively enhances LLM performance, achieving factual, up-to-date, and reasonable outcomes.

# 4 EXPERIMENTS AND RESULTS

In this section, we conduct extensive experiments on widely used open-domain generation tasks to validate the effectiveness of the proposed method in enhancing the factualness of LLM generation.

## 4.1 EXPERIMENTAL SETTINGS

**Datasets and Evaluation Metrics.** We evaluate the effectiveness of our method on three tasks: **(1)** *Short-form generation:* Following prior work (Asai et al., 2023), we evaluate performance on the PopQA (Mallen et al., 2022) and TriviaQA (Joshi et al., 2017) datasets using factual accuracy, which assesses whether the gold answer is included in the model's generated content. We focus on the long-tail subset, consisting of 1,399 queries for PopQA and 11,313 test queries for TriviaQA. **(2)** *Long-form generation task:* we employ ASQA (Stelmakh et al., 2022) and utilize 948 queries in dev set for evaluation. We adopt the official metrics of str-em, Rouge-L (Chin-Yew, 2004) (R-L), QA-Hit (Pillutla et al., 2023), QA-EM, and QA-F1 scores for ASQA. **(3)** *Multi-hop question answering:* we assess accuracy on HotpotQA (Yang et al., 2018) and 2WikiMultiHopQA (2WikiQA) (Ho et al., 2020).

**Baselines.** We utilize Llama3-8B-Instruct (Dubey et al., 2024) as our default generator model. Specifically, we compare our work against two different baselines: (1) **Baseline without Retrieval**, where LLMs generate answers directly without retrieval. Selections involve publicly available models like Llama2-7B, Llama2-13B (Touvron et al., 2023), Llama-8B (Dubey et al., 2024), and private models such as ChatGPT (Ouyang et al., 2022). In addition, to verify the compatibility of our method with various LLMs, we conducted additional simple comparisons with Alpaca (Taori et al., 2023), Vicuna-7B (Zheng et al., 2023) and Mistral-7B (Jiang et al., 2023a). (2) **Baseline with Retrieval**, where an LLM (e.g., Llama2, Llama3) generates output based on the query and top retrieved documents. We consider several advanced RAG methods including Self-RAG (Asai et al., 2023), FLARE (Jiang et al., 2023b), REPLUG (Shi et al., 2023b), SuRe (Kim et al., 2024) and RECOMP (Xu et al., 2024) for comparison. On the same dataset, we consistently use the identical retriever across all experiments to ensure a fair comparison.

| Methods | Short-form | | Multi-hop | | Long-form | | | | |
|---|---|---|---|---|---|---|---|---|---|
| | PopQA | TriviaQA | HotpotQA | 2WikiQA | | | ASQA | | |
| | acc (%) | acc (%) | acc (%) | acc (%) | str-em | R-L | QA-Hit | QA-EM | QA-F1 |
| *Baseline w/o retrieval* | | | | | | | | | |
| ChatGPT* | 29.3 | 74.3 | - | - | 35.3 | - | - | - | - |
| Llama2-7B-Chat | 14.7 | 57.1 | 14.6 | 18.4 | 7.0 | 29.1 | 0.4 | 4.2 | 6.4 |
| Llama2-13B-Chat | 14.7 | 59.3 | 18.7 | 22.3 | 9.2 | 12.4 | 0.8 | 5.2 | 7.9 |
| Llama3-8B-Instruct | 22.8 | 69.4 | 27.7 | 45.6 | 24.4 | 32.8 | 1.8 | 13.4 | 19.5 |
| *Baseline with retrieval* | | | | | | | | | |
| Llama2-7B-Chat | 38.2 | 42.5 | 16.4 | 16.9 | 7.8 | 24.5 | 0.7 | 4.0 | 5.7 |
| Llama2-13B-Chat | 45.7 | 47.0 | 26.3 | 24.5 | 16.7 | 28.1 | 2.9 | 8.9 | 12.6 |
| Llama3-8B-Instruct | 62.7 | 73.0 | 33.6 | 41.3 | 27.6 | 33.8 | 3.6 | 17.3 | 22.9 |
| Self-RAG (Llama2)$_{7B}$ | 52.4 | 66.4 | 27.4 | 35.9 | 30.2 | 35.7 | 3.3 | 18.5 | 24.0 |
| Self-RAG (Llama2)$_{13B}$ | 55.8 | 69.3 | 28.2 | 36.0 | 31.6 | **35.9** | 2.8 | **20.2** | 26.3 |
| Self-RAG (Llama3)$_{8B}$ | 50.2 | 71.4 | 14.9 | 32.9 | 26.7 | 32.8 | 2.3 | 14.4 | 19.5 |
| FLARE | 16.7 | 53.4 | 19.5 | 25.6 | 13.1 | 9.3 | 0.4 | 9.5 | 12.8 |
| REPLUG | 37.4 | 60.8 | 16.2 | 19.9 | 20.9 | 11.2 | 1.1 | 14.7 | 20.2 |
| SuRe | 54.8 | 53.2 | 18.5 | 16.6 | 20.5 | 5.8 | 0.7 | 13.6 | 19.3 |
| RECOMP | 62.8 | 60.2 | 25.2 | 32.0 | 24.4 | 8.0 | 1.3 | 15.0 | 21.1 |
| Ours | **68.5** | **77.0** | **46.0** | **52.8** | **35.0** | 35.2 | **4.0** | 20.1 | **26.9** |

Table 1: State-of-the-art comparison on various open-domain question answering datasets. We re-implement the baselines and report their performance as the maximum value between the original scores and our reproduced results. An asterisk * indicates results copied from (Asai et al., 2023) for reference. A dash "-" denotes results that are either not reported in the original paper or are not applicable. The best performance is highlighted in ***bold***.

**Implementation Details.** For model training, we utilize a small language model Qwen-2-0.5B (Yang et al., 2024a) as the base model of our analyzer to avoid introducing substantial computational demands. Consistent with prior research (Asai et al., 2023), GPT-4 is employed to generate the training data for query decoupling and relevance assessment. Detailed statistics of the training datasets and additional information are provided in the Appendix Section B.

For model inference, following previous work (Asai et al., 2023), we adopt Contriever-MS MARCO (Izacard & Grave, 2020) as the retriever model for PopQA and TriviaQA. For ASQA, HotpotQA, and 2WikiMultiHopQA, we utilize the author-provided retrieval documents across all baselines to ensure a fair comparison. The weighting parameters $\alpha$ and $\beta$ in DRA default to 0.6 and 0.4. For CRC, the initial value of $\gamma$ is set to 3. More details are outlined in the Appendix Section C.

## 4.2 IN-DEPTH RESULTS

**Comparison against Baselines without Retrieval.** Table 1 (top) presents the baselines without retrieval. Our method demonstrated significantly superior performance compared to existing fine-tuned LLMs across all datasets. Notably, on the PopQA dataset, our method achieved a retrieval performance gain of 45.7%. This demonstrates the strong capability of our method in granular retrieval and utilization of relevant information. The performance improvement of our method on certain datasets such as ASQA was less pronounced. This could be attributed to the presence of some relevant information in the training data of these models, diminishing the impact of externally retrieved knowledge.

**Comparison against Baselines with Retrieval.** The bottom of Table 1 presents the performance comparison between our method and baseline with retrieval. Our method significantly outperforms existing RAG approaches across nearly all generation tasks:

**(1)** *Short-form generation:* Our method achieved a 5.7% improvement in accuracy on the PopQA dataset and a 4% improvement on TriviaQA, compared to the next best approach. This demonstrates that our Lexical Diversity-aware RAG can effectively enhance retrieval precision and the utilization

| Self-RAG | Llama3-Instruct-8B | Ours |
|---|---|---|
| *Question: What star sign is Jamie Lee Curtis? [Ground Truth: "Scorpio"]* | | |
| Jamie Lee Curtis is a Cancer. | Jamie Lee Curtis was born on November 22, 1958, which makes her a Sagittarius. | Jamie Lee Curtis is a Scorpio, born on November 22, 1958. |
| *Question: Who was known by his stage name Aladin and helped organizations improve their performance as a consultant? [Ground Truth: "Eenasul Fateh"]* | | |
| James P. Comer | The text that refers to James P. Comer as a consultant is: 
 - James P. Comer 
 This text does not mention a stage name, and James P. Comer is a well-known figure. | Eenasul Fateh, also known by his stage name Aladin, is a Bangladeshi-British cultural practitioner, magician, live artist, and former international management consultant. |

Table 2: Case study on TriviaQA and HotpotQA. Blue text indicates correct output, while red text represents incorrect output.

of relevant information. The case study in Table 2 further illustrates the effectiveness of our method in aggregating relevant information to enhance the generation process. Self-RAG generated content based on incorrect information, while Llama3-Instruct-8B, despite retrieving accurate information, was unable to produce the correct output due to interference from irrelevant noise. Only our method addresses the issues faced by other approaches through granular relevance assessment and calibration, facilitating the factualness of generation.

**(2) *Muti-hop question answering:*** Our method achieves substantial improvements on multi-hop tasks, demonstrating a 12.4% increase in accuracy on HotpotQA and an 11.5% increase on 2Wiki-MultiHopQA. Multi-hop tasks typically involve complex queries that require information retrieval from multiple documents to support accurate generation. These tasks impose higher demands on the capacity for precise knowledge acquisition and utilization, further underscoring the effectiveness of our method's granular relevance assessment. In the second example of HotpotQA in Table 2, only our method accurately retrieved the relevant information and generated the correct output.

**(3) *Long-form generation:*** On the ASQA dataset, the str-em metric, which quantifies the alignment between generated content and ground truth, indicates that our method attained optimal performance, highlighting its precise knowledge extraction and calibration capabilities. The QA-Hit, QA-EM, and QA-F1 metrics offer an objective evaluation of the generated content through a question-answer framework. Our approach demonstrates superior performance in QA-Hit and QA-F1, with only a marginal underperformance in QA-EM compared to Self-RAG. This slight discrepancy may be attributed to the limited presence of irrelevant documents in the official dataset, which may have constrained our model's ability to fully exploit its information assessment capabilities.

## 5 ANALYSIS

In this section, we conduct a comprehensive analysis of our granular relevance assessment based RAG. All experiments are implemented on Llama3-Instruct-8B.

### 5.1 ABLATION STUDY

**Ablation of Modules.** We first conduct ablation experiments on PopQA, TriviaQA, HotpotQA, and 2WikiMultiHopQA to separately investigate DRA and CRC modules in Table 3. The baseline model achieves 33.6% accuracy on HotpotQA and 41.3% on 2WikiMultiHopQA. Simply employing the DRA module will bring huge 11.3% accuracy gains on HotpotQA and 11.5% on 2WikiMulti-HopQA. It reveals that introducing distinct relevance assessment criteria facilitates precise document relevance evaluation, thereby improving the retrieval of relevant information. Additionally, solely deploying the CRC module will bring 3.1% improvement on 2WikiMultiHopQA. This indicates that adjusting generation by eliminating the noisy decoding distribution effectively promotes the aggregation and utilization of relevant information. The absence of Contrastive Relevance Calibration

| DRA | CRC | PopQA | TriviaQA | HotpotQA | 2WikiQA |
|-----|-----|-------|----------|----------|---------|
|     |     | 62.7 | 73.0 | 33.6 | 41.3 |
|     | ✓ | 65.0 (↑**2.3%**) | 73.1 (↑**0.1%**) | 35.2 (↑**1.6%**) | 44.4 (↑**3.1%**) |
| ✓ |     | 64.0 (↑**1.3%**) | 76.4 (↑**3.4%**) | 44.9 (↑**11.3%**) | 51.4 (↑**10.1%**) |
| ✓ | ✓ | **68.5** (↑**5.8%**) | **77.0** (↑**4.0%**) | **46.0** (↑**12.4%**) | **52.8** (↑**11.5%**) |

Table 3: Ablation study on the impact of DRA and CRC. Our full model yields superior performance, and each module contributes to the proposed method.

prevents the model from reaching optimal performance. Therefore, the DRA and CRC modules should work synergistically to fully enhance the overall method.

**Ablation of Hyper-parameter.** We first analyze two parameters that regulate relevance assessment in our DRA module: the weight of variant component scores $\alpha$ and the weight of variant component scores $\beta$. Figure 6 (a) shows the variation in model accuracy on PopQA as parameters $\alpha$ and $\beta$ change. It can be observed that, compared to supplementary components, variable components have a more significant impact on our model's performance. As $\alpha$ increases, the accuracy exhibits an inverted U-shaped trend. Our method reaches its peak performance at $\alpha = 0.6$ and $\beta = 0.4$, achieving an accuracy of 68.5% on PopQA. Additionally, we validated the effect of calibration degree $\gamma$ in the CRC module on model accuracy, as shown in Figure 6 (b). The model's performance steadily improves as the calibration degree increases, with the rate of improvement gradually leveling off. This further demonstrates that our contrastive relevance calibration effectively enhances the factual accuracy of the model's generation.

## 5.2 DEEP ANALYSIS

**Influence of Training Data Size.** We analyze the impact of DRA's training data on model accuracy using the PopQA dataset. The training data consists of two parts: data for query decomposition (shown in Figure 3) and data for relevance evaluation (shown in Figure 4). The results show that model performance gradually improves as the amount of two kinds of training data increases, with significant performance gains achieved even with a relatively small dataset. It can be observed that for data related to query decomposition, only around 1,000 samples are needed to achieve significant performance improvement. Similarly, for data related to relevance evaluation, fewer than 5,000 samples are sufficient to realize a substantial performance boost. This highlights the efficiency and low resource demands of our approach. The slight performance drops in the curve may be attributed to domain differences between the training data and PopQA, which could be further explored in future work by increasing the diversity of the training data.

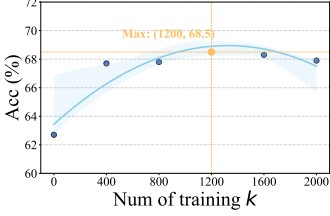 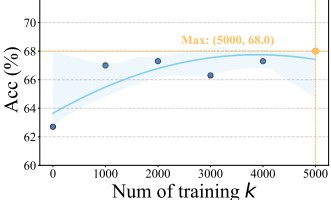 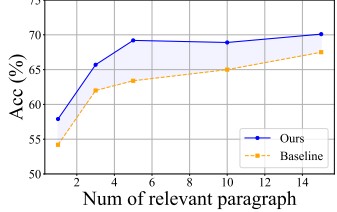

Figure 3: Influence of query decomposition data.

Figure 4: Influence of relevance assessment data.

Figure 5: Accuracy with relevant parameter data

**Influence of Retrieved Document Volume.** We conduct an analysis of how the number of retrieved documents affects the model's performance. Figure 5 compares the outputs of our method with those of the Llama3 retrieval-based approach across different numbers of retrieved documents. The results demonstrate that our method consistently outperforms the baseline with retrieval in all scenarios. Initially, the relative performance improvement increases as more documents are retrieved. This improvement can be attributed to our method's superior evaluation of document rele-

vance, which ensures that the retrieved documents are genuinely relevant to the query, thereby enhancing model performance. The performance gain reaches its maximum when five documents are retrieved, after which a gradual decline is observed. This decline occurs because, beyond a certain threshold, the proportion of truly relevant documents decreases as more documents are retrieved, resulting in diminishing returns. Nonetheless, our method continues to deliver positive performance improvements, even as the number of retrieved documents increases.

| Methods | PopQA | TriviaQA |
|---------|-------|----------|
| Llama3$_{8B}$ | 62.7 | 73.0 |
| Ours w/o decoding | 64.0 | 76.4 |
| Ours w/o irrelevant docs | 66.5 | 76.1 |
| Ours w/ fixed irrelevant doc | 67.8 | 75.8 |
| Ours | **68.5** | **77.0** |

Table 4: Impact of decoding strategy on performance.

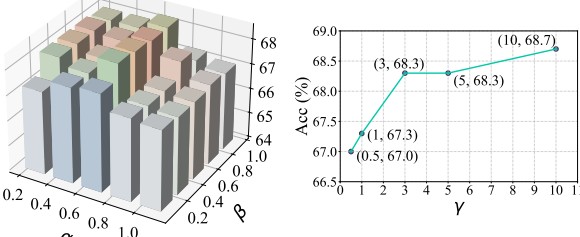

Figure 6: Analysis of DRA and CRC hyper-parameters.

**Influence of Noise Reference.** To further validate the effectiveness of our CRC, we compared several proposed strategies for constructing irrelevant noise references and contrasted them with a noise-free distribution decoding calibration setting. Table 4 presents the accuracy of different irrelevant noise reference construction strategies on PopQA. It can be observed that the performance of our noise strategies outperforms the noise-free decoding calibration setting, indicating that our Contrastive Relevance Calibration effectively reduces the impact of irrelevant information and improves the factual accuracy of the model's output. Additionally, we selected a fixed irrelevant document as noise, which bears almost no similarity to the query. It can be observed that the performance under our noise reference construction strategy is superior. This suggests that strongly interfering noise documents better simulate the irrelevant noise found in retrieved documents under real-world conditions, leading to more effective calibration of the model's generation.

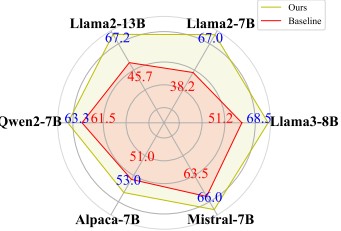

Figure 7: Our method on different baselines.

**Influence of Different LLMs.** To further validate the adaptability of our method to other LLMs, we select other representative fine-tuned LLMs as generator models and conduct experiments on PopQA. As shown in Figure 7, the results demonstrate that our method outperforms the baseline for all LLMs. It is worth mentioning that our method has significantly improved the Llama models, notably boosting the accuracy of Llama2-7B-Chat from 38.2% to 67.0%. It confirms that our method is compatible with various LLMs and can effectively enhance their performance.

## 6 CONCLUSION

In this paper, we introduced Lexical Diversity-aware RAG, a retrieval-augmented generation framework designed to address the limitations of existing RAG methods by incorporating granular relevance assessment. The Diversity-sensitive Relevance Analyzer enhances the precision of retrieval by applying tailored criteria based on lexical diversity, while the Contrastive Relevance Calibration Module refines the generation process by mitigating the impact of irrelevant information through contrastive calibration. Our approach effectively improves the retrieval of semantically aligned documents and promotes the aggregation of relevant knowledge, leading to significant advancements in LLM generation. The results from extensive evaluations on several open-domain question-answering benchmarks validate the superiority of our method, demonstrating its potential to substantially enhance factual accuracy in a wide range of applications. A potential limitation of our method is its generalization to more complex generation tasks, such as multi-turn dialogue. In the future, we will further explore the application of Lexical Diversity-aware RAG across different domains.

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

## A  MORE RELATED WORK

**Contrastive Decoding.**    Contrastive decoding (Li et al., 2022) is a technique for enhancing open-ended text generation without requiring additional training, achieved by maximizing the difference in log probabilities between an expert LLM and an amateur LLM. This method has demonstrated strong performance across various domains, including reasoning (O'Brien & Lewis, 2023) and neural machine translation (Waldendorf et al., 2024). CAD (Shi et al., 2023a) employs pointwise mutual information to adjust the output probability distribution, addressing conflicts between internal and external knowledge within the model. Additionally, Chuang et al. (2023) propose a decoding strategy that contrasts different layers of the same LLM to more effectively highlight factual knowledge acquired during pre-training. In contrast to these contrastive decoding approaches, our Contrastive Relevance Calibration refines the model's generation by calibrating it against a constructed noise reference, reducing the impact of granular irrelevant text and enhancing the model's ability to aggregate and utilize relevant knowledge.

## B  MORE DETAILS OF DRA

**Inference of DRA.**    The DRA module performs a granular relevance assessment on the documents returned by the retriever during the retrieval to accurately identify the documents truly relevant to the query. Specifically, in the retrieval stage, given a query $x$ and several retrieved documents $\mathbf{D}$ return by embedding-based retriever, we train the DRA module $\mathcal{R}$ based on a lightweight language model to conduct a further assessment of the intrinsic relevance between each document $d_i \in \mathbf{D}$ and $x$. We first prompt the DRA module $\mathcal{R}$ with the instruction $\mathrm{I_{dec}}$ to decouple the query $x$ into various components $\mathbf{C} = \{< c_1, a_1 >, < c_2, a_2 >, \ldots, < c_n, a_n >\}$. Each component $c_j$ is attributed as $a_j$ based on its lexical diversity and extraction strategy during decoupling. We predefined three different component attributes and established progressive assessment criteria based on their lexical diversity. Then we prompt the DRA module with instruction $\mathrm{I_{sco}}$ to analyze the relevance of the retrieval document $d_i \in \mathbf{D}$ and each component $c_j$ and assign a relevance score $s_{ij}$. All component scores are integrated to accurately characterize the overall relevance of the document $d_i$ to the entire query as $s_i$. We then sort documents and sent the top $r$ retrieved documents $\mathbf{D}^{\mathrm{rel}} = \{d_1^{'}, d_2^{'}, \ldots, d_r^{'}\}$ according to their relevance to model $\mathcal{M}$ for enhancing generation.

**Evaluation Criteria.**    For the evaluation criteria $\sigma_1$, a score of 1 is assigned if the document explicitly matches the extracted invariant component; otherwise, it is set to 0. As for the evaluation criteria $\sigma_2$, the score is a continuous value between [0,1], reflecting the document's relevance to both the variant and supplementary components of the query. A higher score indicates stronger relevance to the variant component, while the supplementary component is evaluated more leniently, requiring only partial relevance to achieve a score within the same range.

**Data Collection for DRA.**    The instance $(\mathbf{i}, \mathbf{o})$ of the DRA training data consists of two different types: (1) Query decomposition data. The input of DRA $\mathbf{i}$ is the query $\mathbf{x}$ and the instruction $\mathrm{I_{dec}}$ (*"You are an assistant in extracting key components from a given question."*), the output $\mathbf{o}$ is the decoupled components set $C$ extracted based on $\mathbf{x}$; (2) relevance assessment data. The input $\mathbf{i}$ is a combination of the query $\mathbf{x}$, the retrieved document $d$, the decoupled components set $C$, and the output $\mathbf{o}$ is an analysis of the relevance between the retrieved document $d$ and each component in $C$, including both match scores and explanations. Following the approach of works (Asai et al., 2023), we utilize the state-of-the-art LLM GPT-4 to generate training data for both the components set construction and the relevance assessment processes. Specifically, we prompt GPT-4 with type-specific instructions followed by few-shot demonstrations of the original task input $\mathbf{x}$ to generate the decoupling components set $C$. Next, we prompt GPT-4 with instructions followed by few-shot demonstrations of the original task input $\mathbf{x}$, the generated components set $C$, and the retrieved documents $D$ to predict the semantic matching analysis. Manual evaluation indicates that GPT-4's predictions align well with human assessments. We collected a total of 1200 instances for decoupling components set construction data and 5543 instances for progressive relevance assessment data to form the supervised training dataset for the analyzer.

Table 5 presents the sources and statistics of our training data. Specifically, considering the characteristics of open-domain question-answering tasks, we select a subset of data from the PopQA and

TriviaQA training sets without losing generality. For each query, we employ type-specific instructions accompanied by 2-3 example prompts to guide GPT-4 in generating component decoupling data. Subsequently, we use Contriever as the retriever to obtain 10 documents similar to the query and prompt GPT-4 to generate relevance analysis results for each document and the query's decoupled components. The prompt and examples for generating component decoupling data are shown in Table 10, while the prompt and examples for generating relevance analysis results and scores are presented in Table 11.

| Data Type | Training Sample Size | Data Source | Selected Sample Size from Source |
|---|---|---|---|
| Component Extraction | 1200 | PopQA Training Set | 2100 |
| | | TriviaQA Training Set | 1756 |
| Consistency Evaluation | 5543 | PopQA Training Set | 1990 |
| | | TriviaQA Training Set | 4553 |

Table 5: The sources and statistics of training data.

**DRA Learning.** We employ the commonly used cross-entropy loss for supervised fine-tuning of the analyzer:

$$\mathcal{G}_\theta = \arg\min_\theta \mathbb{E}_{(\mathbf{i},\mathbf{o})\sim\mathcal{D}}[\text{CE}(\mathcal{G}(\mathbf{o};\theta),\mathbf{i})] \tag{7}$$

Where $\theta$ denotes the learned parameter of $g$ and CE refers to the cross-entropy loss.

**Details of Training Settings.** To minimize additional computational overhead during inference, we employed instruction fine-tuning to train a small language model Qwen-2-0.5B (Yang et al., 2024a) as our DRA module. This training process requires only a small amount of data and computational resources.

## C  EXPERIMENTAL DETAILS

**More Implementation.** By default, we assess relevance using 10 documents per query and select the top 5 for augmentation during model generation. For PopQA and TriviaQA, we follow prior work by using Wikipedia as the retrieval corpus. For HotpotQA, 2WikiQA, and ASQA, we use the official retrieval documents provided by each dataset to ensure fair comparisons.

| Methods | PopQA |
|---|---|
| Llama3$_{8B}$ | 62.7 |
| monoBERT | 66.0 |
| monoT5 | 66.5 |
| Ours | **68.5** |

Table 6: Performance comparison of reranking strategies.

**Comparison with RAG Rerankers.** To further validate the effectiveness of our DRA module in assessing the relevance of retrieved documents, we compared it with embedding-based retrieval reranking methods. We selected two typical rerankers, monoBERT and MonoT5, replacing our DRA module with these methods, and applied the CRC based on the resulting re-ranking. Table 6 presents the experimental results on PopQA, where our approach significantly outperformed both rerankers in terms of accuracy. This further supports our motivation: calculating similarity between the entire query and the retrieved documents to represent relevance is inherently biased, whereas our DRA reasoning and analysis enable a more accurate assessment of relevance.

**Effectiveness of individual components within DRA.** We conducted a more detailed analysis of the contributions of each category of components within DRA. The results in Table 7 demonstrate that each category contributes to the overall performance of our method. Notably, the performance contributions of the invariant, variant, and supplementary components decrease sequentially. This observation aligns with our hypothesis that the importance of these three categories for relevance assessment diminishes in the same order. We will follow your suggestion and include this detailed ablation experiment and analysis in the paper.

**Generalization of DRAG**   we conducted validation on the FreshQA dataset, a non-Wikipedia style dataset, as shown in Table 8. The results indicate that our method delivers significant performance improvements on FreshQA, significantly surpassing the baseline methods. This highlights the strong generalization capabilities of our approach across different datasets. In fact, the non-Wikipedia style of a dataset does not significantly affect the performance of DRAG. This is because DRAG primarily focuses on decoupling the query within the dataset, and as long as the different components of the query can be accurately identified, DRAG can function effectively.

**The Training efficiency of DRAG**   A significant advantage of our approach is its ability to deliver substantial performance gains without requiring extensive training resources or time. Table 9 compares the data size and training cost of our method with the training-based RAG approach (Self-RAG), demonstrating the low resource consumption of our method.

| Methods | PopQA |
|---|---|
| LLama3-RAG | 62.7 |
| w/o DRA | 65.0 |
| 0.9 Invariant only | 67.9 |
| Variant only | 66.2 |
| 0.9 Supplementary only | 65.6 |
| Ours | **68.5** |

Table 7: Effectiveness of individual components in DRA. Each category contributes to the overall performance of our method.

| Methods | ACC (%) |
|---|---|
| Llama3-8B with retrieval | 23.0 |
| Self-RAG | 20.2 |
| Ours | **26.2** |

Table 8: Accuracy comparison on FreshQA.

| Methods | Number of Instances | Model Size | Training Cost |
|---|---|---|---|
| Self-RAG | 145,619 | 7B | 4 Nvidia A100 with 80GB memory |
| Ours | **6,743** | **0.5B** | **1 Nvidia V100 GPU with 24G memory** |

Table 9: Comparison of number of instances, model size, and training costs.

# D   INSTANCES OF TRAINING DATA

In this section, we provide 10 examples of query decoupling data and relevance assessment data in the appendix. From these samples, it is evident that our generated data is generally accurate and reasonable, effectively capturing the logical reasoning processes involved in query decoupling and relevance assessment.

---

**input:**  What sport does Nobuhiro Ishizaki play?
**output:**  Nobuhiro Ishizaki (invariant); Sport (variant).
**system:**  You are an assistant in extracting key elements from a given question.

---

**input:**  What was Cleopatra's favourite colour?
**output:**  Cleopatra (invariant); favourite colour (variant).
**system:**  You are an assistant in extracting key elements from a given question.

---

**input:**  What disease claimed the life of notorious American gangster, Al Capone?
**output:**   Al Capone (invariant); disease claimed the life (variant); notorious American gangster (supplementary)
**system:**  You are an assistant in extracting key elements from a given question.

**input:** The Torres Strait lies between Australia and which island?
**output:** The Torres Strait (invariant); lies between Australia (variant); which island (initial)
**system:** You are an assistant in extracting key elements from a given question.

---

**input:** In Britain, by an act of 1541 (repealed in 1845) artificers, labourers, apprentices and servants were forbidden to play which sport at any time except Christmas, and then only in their master's house and presence, or be fined 6s. 8d?
**output:** Britain, 1541 act (invariant); forbidden sport, Christmas, master's house and presence (variant); artificers, labourers, apprentices, servants (supplementary)
**system:** You are an assistant in extracting key elements from a given question.

---

**input:** Question: What sport does Roland Zajmi play?
Element: Roland Zajmi (invariant); Sport (variant).
Paragraph: Roland Zajmi (born 6 November 1973 in Tiranë) is an Albanian retired footballer who played the majority of his career in Greece with Proodeftiki, Atromitos, Kassandra, Agios Dimitrios and Pierikos. He also played for Dinamo/Olimpik Tirana and Partizani Tirana in Albania, Apollon Limassol in Cyrpus as well as the Albania national team.
**output:** Roland Zajmi (invariant): 1.0 (Roland Zajmi is explicitly mentioned)
Sport (variant): 1.0 (Football is a sport and it's well detailed)
**system:** You are an assistant in scoring paragraphs based on a given question and its associated elements.

---

**input:** Question: What is Quebec City the capital of?
Element: Quebec City (invariant); Capital (variant); Province (supplementary).
Paragraph: Culture of Quebec City, the provincial capital (albeit dubbed La capitale nationale, national capital, in French), is best known as the first permanent settlement and the only fortified city in North America north of Mexico. The old city, partially encircled within the centuries-old walls, has a European flair.
**output:** Quebec City (essential): 1.0 (Quebec City is explicitly mentioned) Capital (initial): 1.0 (Provincial capital is related to capital) Province (supplementary): 0.0 (Province is not mentioned) **system:** You are an assistant in scoring paragraphs based on a given question and its associated elements.

---

**input:** Question: Who is the author of Occupy?
Element: Occupy (invariant); Author (variant).
Paragraph: Occupy (book) The book includes an editor's note, a brief section providing legal advice for American Occupy activists, and five sections written by Chomsky himself. Occupy opens with an editor's note written by Greg Ruggiero, in which he explains the basics to Chomsky's views on the Occupy movement, drawing quotes from his various public speeches in order to do so. Ruggiero also discusses Occupy's success in the United States, stating that it has helped to change media discussions by introducing terms like "the 99%" into popular discourse and also by bringing national attention to the plight of the impoverished. He remarks that the protest movement has."
**output:** Occupy (essential): 1.0 (Occupy is explicitly mentioned)
Author (initial): 0.8 (Author Chomsky is mentioned but not directly as the author)
**system:** You are an assistant in scoring paragraphs based on a given question and its associated elements.

**input:** Question: Rimini Miramare airport has been renamed in honour of which noted film director?
Element: Rimini Miramare airport (invariant); has been renamed in honour of (variant); noted film director (variant).
Paragraph: Federico Fellini by Nino Rota during the funeral ceremony. Five months later, on 23 March 1994, Fellini's widow, actress Giulietta Masina died of lung cancer. Fellini, Masina and their son, Pierfederico, are buried in a bronze sepulchre sculpted by Arnaldo Pomodoro. Designed as a ship's prow, the tomb is located at the main entrance to the Cemetery of Rimini. The Federico Fellini Airport in Rimini is named in his honour. Fellini was raised in a Roman Catholic family and considered himself a Catholic. As an adult, however, he avoided formal activity in the Catholic Church. Films by Fellini included Catholic themes;
**output:** Rimini Miramare airport (invariant): 1.0 (Rimini Miramare airport is explicitly mentioned as Ṫhe Federico Fellini Airport in Rimini) has been renamed in honour of (variant): 0.6 (The paragraph does not explicitly mention the renaming, but it implies it by stating the airport in Rimini is named after Fellini) noted film director (variant): 1.0 (The paragraph mentions that the airport is named after Italian filmmaker Federico Fellini) No supplementary element provided.
**system:** You are an assistant in scoring paragraphs based on a given question and its associated elements.

**input:** Question: What is Delhi the capital of?
Element: Delhi (invariant); Capital (variant); Country (supplementary).
Paragraph: Delhi Delhi, officially known as the National Capital Territory (NCT) of Delhi, is a city and a union territory of India containing New Delhi, the capital of India. It is bordered by the state of Haryana on three sides and by Uttar Pradesh to the east. The NCT covers an area of 1484 km2. According to the 2011 census, Delhi's city proper population was over 11 million, the second-highest in India after Mumbai, while the whole NCT's population was about 16.8 million. Delhi's urban area is now considered to extend beyond the NCT boundaries, and include the neighbouring satellite cities of Ghaziabad, Faridabad, Gurgaon and Noida in an area called the National Capital.
**output:** Delhi (invariant): 1.0 (Delhi is explicitly mentioned) Capital (variant): 1.0 (Capital is explicitly mentioned and discussed) Country (supplementary): 1.0 (Country, India, is explicitly mentioned)" **system:** You are an assistant in scoring paragraphs based on a given question and its associated elements.

**Prompt for GPT-4.**

You will receive a series of instructions but do not answer them directly. Follow these steps:

1. For each instruction, extract key elements for retrieval from the instruction. Each element should be a word or a phrase. In the output, separate each element extracted from the same instruction with a semicolon.

2. Categorize the extracted elements into three types and annotate them in parentheses after every extracted element in the output:

- Invariant elements: Explicitly extracted from the instruction without any changes. Subsequent verification of the retrieved document will require that the retrieved document explicitly contain Invariant elements. Therefore, the extracted Invariant elements must be able to represent the most basic information in the original instruction, such as personal names, place names, and other entities. Limit the number of invariant elements to a maximum of 1.

- Variant elements: Explicitly extracted from the instruction, but the retrieved documents don't need to explicitly include these elements, just be implicitly related to them. No quantity limit. Ensure the combination of all Invariant and Variant elements fully represents the original instruction.

- Supplementary elements: Inferred based on the instruction's context to clarify the search, not explicitly mentioned, and the retrieved documents don't need to explicitly include them, just be implicitly related to them. Supplementary elements are not necessary and should be minimized in number. They can only be added if the combination of Invariant and Variant elements is not clear when used for retrieval.

Finally, list the key elements separately for each instruction, indicating their category (invariant, variant, or supplementary).

Example and output format:

Example 1:

Instruction: FDA gives fast track status to AstraZeneca's diabetes drug Farxiga.

Elements: Farxiga (invariant); Fast track status (variant); FDA (variant); AstraZeneca (variant); Diabetes drug (variant); Drug approval (supplementary); Regulatory process (supplementary); Pharmaceuticals (supplementary);

Example 2:

Instruction: Does a surgical mask help avoid COVID-19?

Elements: COVID-19 (invariant); Surgical mask (variant); Help avoid (variant)

**Instances of training data.**

**instruction:** Question:

**input:** What is Delhi the capital of?

**output:** Delhi (invariant); Capital (variant); Country (supplementary).

**system:**

You are an assistant in extracting key elements from a given question. The types of elements are defined as follows:

1. Invariant: Elements explicitly mentioned in the question that must be found in the document.

2. Variant: Elements explicitly mentioned in the question that may not be directly found in the document, but the document should contain related information.

3. Supplementary: Elements implied by the question that should be inferred based on the document's content.

Table 10: Prompt instructions for extracting key elements from instructions and categorizing them as invariant, variant, or supplementary.

**Prompt for GPT-4.**

You are tasked with evaluating the relevance of some given paragraphs to a specific question based on the following elements: invariant, variant, and supplementary.

Scoring Standard:

Score of Invariant Element: Check if the invariant element is explicitly mentioned in the paragraph. If it is, assign an invariant score of 1.0; otherwise, assign a score of 0.0.

Score of Variant Element or Supplementary Element: Consider how well the paragraph discusses or relates to the concept or entity represented by the variant element or supplementary element. Assign a variant score ranging from 0 to 1, where 1.0 indicates a strong relevance and 0 indicates no relevance.

Output Format:

paragraph_id: <ID>

invariant_score: <Score>

variant_score: <Score>

supplementary_score: <Score>

Example:

Question: What is Henry Feilden's occupation? Elements: Henry Feilden (Invariant); Occupation (Variant)

Paragraphs: { id: 11341299, title: Henry Feilden (Conservative politician), text: Henry Master Feilden (21 February 1818 – 5 September 1875) was an English Conservative Party politician. }

Output:

id: 11341299. Invariant Score: 1.0 (Henry Feilden is explicitly mentioned). Variant Score: 0.8 (Politician is related to occupation, but not fully detailed). Supplementary Score: 0.0 (No supplementary element provided)

**Instances of training data.**

**instruction:**

Question:

Elements;

Paragraphs:

**input:**

### Question: What was Shoemaker-Levy 9, that crashed into Jupiter in 1994?

### Element: Shoemaker-Levy 9 (invariant); crashed into Jupiter in 1994 (variant)

### Paragraph: Carolyn S. Shoemaker

of Technology, Pasadena, California, and the Palomar Observatory, San Diego, California. That year, Shoemaker was hired at the United States Geological Survey as a visiting scientist in the astronomy branch, and then in 1989 began work as an astronomy research professor at Northern Arizona University. She concentrated her work on searching for comets and planet-crossing asteroids. Teamed with astronomer David H. Levy, the Shoemakers identified Shoemaker-Levy 9, a fragmented comet orbiting the planet Jupiter on March 24, 1993. After Gene's death in 1997, Shoemaker continued to work at the Lowell Observatory with Levy, and continues to work there today. In

**output:**

Shoemaker-Levy 9 (invariant): 1.0 (Shoemaker-Levy 9 is explicitly mentioned)

crashed into Jupiter in 1994 (variant): 0.6 (The paragraph mentions the discovery of the comet and its orbit around Jupiter, but does not explicitly mention the collision in 1994)

No supplementary element provided

**system:**

You are an assistant in scoring paragraphs based on a given question and its associated elements. Scoring rules:

1. Invariant Score: If the invariant element is explicitly mentioned in the paragraph, assign an invariant score of 1.0; otherwise, assign a score of 0.0.

2. Variant Score and Supplementary Score: Consider how well the paragraph relates to the variant element or supplementary element. Assign a variant score ranging from 0 to 1, where 1.0 indicates strong relevance and 0 indicates no relevance.

Table 11: Prompt instructions for evaluating paragraph relevance based on invariant, variant, and supplementary elements.

# E    MORE CASE STUDY.

We present a case study illustrating how our method addresses the challenge of lexical diversity in Table 12. In the example, a query and two relevant document passages are analyzed. The baseline retrieval fails to maintain the lexical integrity of "Arcangelo Ghisleri," instead being misled by repeated occurrences of "Arcangelo" in unrelated contexts, resulting in irrelevant documents. Our method, by analyzing lexical diversity and applying refined evaluation criteria, identifies the passage that exactly matches "Arcangelo Ghisleri" and implicitly references "occupation" through the term "journalist." This ensures the retrieval of the correct document, enabling accurate model outputs.

---

**Question:** What is Kyaw Swe's occupation?

**Retrieval Document (Baseline):**

Paragraph 1: S. Michele Arcangelo, archangel in Jewish, Christian, and Islamic teachings ; Andrea di Cione Arcangelo (1308–1368), Italian painter, sculptor, and architect active in Florence ; Antonio di Arcangelo, Italian painter, active in Florence in a Renaissance style, between 1520 and 1538 ; Arcangelo Califano (1730–1750), baroque composer and cellist ; Arcangelo Placenza da Calatafimi, (1390–1460) venerated Italian Franciscan friar and preacher ; Arcangelo Canetoli (1460–1513), venerated Catholic priest ; Arcangelo Cascieri (1902–1997), influential sculptor, major figure in Boston Architectural College in Boston, Massachusetts ; Arcangelo di Cola (active 1416-1429) Italian late-Gothic painter ; Arcangelo Corelli (1653–1713), Italian violinist and composer of Baroque music ; Arcangelo Ghisleri (1855–1938), geographer who created numerous maps of Africa ; Arcangelo Guglielmelli (c. 1650–1723), Italian

Paragraph 2: Arcangelo Guglielmelli (c.1650—1723) was an Italian architect and painter, active in his native Naples, Italy, in a late-Baroque style. He was involved in the building and reconstruction of churches, many of which had been damaged by the earthquakes of 1688 and 1694.

**Answer (Baseline):**   Arcangelo Ghisleri is a geographer who created numerous maps of Africa.

- - - - - - - - - - - - - - - - - - - - - - - - - - - - - - - - - - - - - - - - - - -

**Retrieval Document (Ours):**

Paragraph 1: Ghislieri ; Michele Ghislieri (1504–1572), also known as Pope Pius V. ; Ghislieri College ; Ghislieri Choir and Consort, Giulio Prandi Ghisleri ; **Arcangelo Ghisleri (1855–1938), an Italian journalist.** The Ghislieri, or less commonly Ghisleri, were an ancient Bolognese aristocratic family.

Paragraph 2: S. Michele Arcangelo, archangel in Jewish, Christian, and Islamic teachings ; Andrea di Cione Arcangelo (1308–1368), Italian painter, sculptor, and architect active in Florence ; Antonio di Arcangelo, Italian painter, active in Florence in a Renaissance style, between 1520 and 1538 ; Arcangelo Califano (1730–1750), baroque composer and cellist ; Arcangelo Placenza da Calatafimi, (1390–1460) venerated Italian Franciscan friar and preacher ; Arcangelo Canetoli (1460–1513), venerated Catholic priest ; Arcangelo Cascieri (1902–1997), influential sculptor, major figure in Boston Architectural College in Boston, Massachusetts ; Arcangelo di Cola (active 1416-1429) Italian late-Gothic painter ; Arcangelo Corelli (1653–1713), Italian violinist and composer of Baroque music ; Arcangelo Ghisleri (1855–1938), geographer who created numerous maps of Africa ; Arcangelo Guglielmelli (c. 1650–1723), Italian

**Answer (Ours):** Arcangelo Ghisleri (1855–1938) is an Italian journalist.

---

Table 12: An example of Llama3-8B and our method on PopQA. The Case illustrates that our method addresses the challenge of lexical diversity.

Additionally, we provide further results on the PopQA, TriviaQA, HotpotQA, 2WikiMultiHopQA, and ASQA datasets. Additionally, we include an example of an irrelevant paragraph along with our corresponding result.

**Question:** What is Kyaw Swe's occupation?
**Self-RAG:** Kyaw Swe is a Burmese politician who currently serves as a House of Nationalities member of parliament for Magway Region No. 11 constituency.
**LLaMA-3:** Kyaw Swe has had multiple occupations throughout his career. Initially, he was a school teacher. Later, he became a politician and served as a member of the National League for Democracy Party. He was elected as a representative in the Amyotha Hluttaw from Magway Region No. 11 parliamentary constituency in the 2015 Myanmar general election.
**Ours:** There are five individuals with the name Kyaw Swe:
Politician 1: Kyaw Tint Swe (born 19 March 1945) He is a former Minister for the Office of the State Counsellor of Myanmar.
Teacher/Politician 2: Kyaw Swe (born 1 December 1960) He graduated as a school teacher and is also a politician.
Politician 3: Kyaw Swe (born 1 December 1979) He is a House of Nationalities member of parliament for Magway Region No. 11 constituency and a member of the National League for Democracy.
Minister: Lt. General Kyaw Swe (no birthdate mentioned) He was the former Minister of Home Affairs of Myanmar, in office from 2016 to 2020, and previously served as Commander of South-west Command in Pathein, Ayeyarwady Region.
Actor: Kyaw Swe (10 February 1924 – 15 August 1982) He was a Burmese actor and film director.
These individuals have distinct occupations and dates of birth, indicating that they are not the same person.

Table 13: Examples of self-RAG, Llama3-8B, and our method on PopQA.

**Question:** Who was the man behind The Chipmunks?
**Self-RAG:** The Chipmunks were created by Ross Bagdasarian Sr.
**LLaMA-3:** The man behind The Chipmunks was Ross Bagdasarian Sr., also known as David Seville. He was an American singer-songwriter and actor who created the novelty musical group The Chipmunks (now better known as "Alvin and the Chipmunks"). He produced and directed "The Alvin Show", which aired on CBS in 1961-62, and wrote and recorded many of the group's hit songs, including "Witch Doctor" and "The Chipmunk Song (Christmas Don't Be Late)". Bagdasarian died of a heart attack in 1972, bringing all Chipmunk activity to an abrupt halt. His legacy lives on through the beloved characters he created.
**Ours:** Ross Bagdasarian Sr. was the man behind The Chipmunks. He was an American singer-songwriter and actor who created the novelty musical group in 1958. Initially a stage and film actor, he rose to prominence with the songs "Witch Doctor" and "The Chipmunk Song (Christmas Don't Be Late)", which both became "Billboard" number-one singles. He produced and directed "The Alvin Show", which aired on CBS in 1961–62. He created the group consisting of three singing animated anthropomorphic chipmunks: Alvin, Simon, and Theodore, managed by their human adoptive father, David (Dave) Seville, who was actually Bagdasarian's stage name. The group was initially known as David Seville and the Chipmunks or simply the Chipmunks before being renamed Alvin and the Chipmunks.

Table 14: Examples of self-RAG, Llama3-8B, and our method on TriviaQA.

**Question:** What distinction is held by the former NBA player who was a member of the Charlotte Hornets during their 1992-93 season and was head coach for the WNBA team Charlotte Sting?

**Self-RAG:** Muggsy Bogues

**LLaMA-3:** Muggsy Bogues, a former NBA player, was a member of the Charlotte Hornets during their 1992-93 season and later became the head coach of the WNBA team Charlotte Sting.

The texts can be categorized into two groups: those related to the Charlotte Hornets (NBA) and those related to the Charlotte Sting (WNBA). The instruction requires finding the connection between the Charlotte Hornets and the Charlotte Sting, which is Muggsy Bogues.

**Ours:** Tyrone Curtis "Muggsy" Bogues, the shortest player ever to play in the National Basketball Association, was a member of the Charlotte Hornets during their 1992-93 season and later served as head coach of the WNBA team Charlotte Sting.

Table 15: Examples of self-RAG, Llama3-8B, and our method on HotpotQA.

**Question:** Who is the mother of the director of film Polish-Russian War (Film)?

**Self-RAG:** The mother of the director of the film "Polish-Russian War" is not publicly known.

**LLaMA-3:** The texts do not mention the mother of the director of the film Polish-Russian War. However, the film Polish-Russian War is mentioned in the provided texts, but it is not clear who the director is.

The texts mention the following individuals:

- Maheen Khan, a Pakistani fashion and costume designer - Viktor Yeliseyev, a Russian general, orchestra conductor, and music teacher - Alice Washburn, an American stage and film actress - Minamoto no Chikako, the mother of Prince Morinaga

None of these individuals are mentioned as the director of the film Polish-Russian

**Ours:** Małgorzata Braunek is the mother of Xawery Żuławski, the director of the film Polish-Russian War (Wojna polsko-ruska).

The relevant texts that refer to Xawery Żuławski's family are:

Xawery Żuławski (born 22 December 1971 in Warsaw) is a Polish film director. In 1995 he graduated National Film School in Łódź. He is the son of actress Małgorzata Braunek and director Andrzej Żuławski.

This text explicitly states that Xawery Żuławski is the son of Małgorzata Braunek and director Andrzej Żuławski.

Małgorzata Braunek is an actress, which is mentioned in the same paragraph as Xawery Żuławski's biography. There is no other text that refers to another Maheen Khan or Alice Washburn being related to Xawery Żuławski or his film Polish-Russian War (Wojna polsko-ruska).

Table 16: Examples of self-RAG, Llama3-8B, and our method on 2WikiQA.

**Question:** The first foreigner to receive bharat ratna was?
**Self-RAG:** The first foreigner to receive the Bharat Ratna was Dr. C. V. Raman, an Indian physicist, in 1954.
**LLaMA-3:** The first foreigner to receive the Bharat Ratna was Khan Abdul Ghaffar Khan, a Pakistani national, who was awarded in 1987.
**Ours:** The first foreigner to receive the Bharat Ratna was Khan Abdul Ghaffar Khan, a Pakistani national, in 1987.

Table 17: Examples of self-RAG, Llama3-8B, and our method on ASQA.

**Question:** What is Richard Harris's occupation?
**Irrelevant Para:** J. Richard Harris
John Richard Harris (born 1910) was an Irish entomologist, fishing consultant, tackle merchant, and author. He was a keen angler and tier of flies from boyhood. He was a sometime merchant seaman, journalist, and freshwater biologist. He was a demonstrator in limnology at Trinity College, Dublin. He was a director of Garnetts & Keegan's Ltd, Dublin, gunsmiths and suppliers of fishing tackle, until his retirement in 1984. He wrote An Angler's Entomology, a book about mayflies for fly fishermen. He has been called, "perhaps the greatest living Irish angler-entomologist". He has also been described as, "a large affable man 'with a sharing attitude towards his whiskey and a colourful manner of expressing his trenchant views on fishing, fishermen, journalists, rugby, life and other matters'
**LLaMA-3:** Richard Harris has multiple occupations.
Richard Harris is a television writer, primarily writing for the crime and detective genres, and has worked on various TV shows and stage productions. He is also a cave diver with over 30 years of experience, having led expeditions and made significant discoveries. Additionally, Richard Harris is an anaesthetist, having completed his medical training and worked as an aeromedical consultant and anaesthetist for the South Australian Ambulance Service. He is also known for his role in the Tham Luang cave rescue and was jointly awarded 2019 Australian of the Year.
There are four different individuals with the name Richard Harris: a television writer, a cave diver and anaesthetist, a former professional footballer, and another anaesthetist who is also a cave diver.
**Ours:** There are four individuals named Richard Harris with different occupations: television writer, footballer, anaesthetist/cave diver, and composer.
Television Writer Richard Harris is a British writer most active from the early 1960s to the mid-1990s, contributing episodes of crime and detective genres and winning the prize for best situation comedy from the Writers' Guild of Great Britain for Outside Edge.
Footballer Richard Harris played in the Football League for Crystal Palace and Wycombe Wanderers during the late 1990s and early 2000s and made an appearance in the FA Youth Cup final against Leeds United.
Two Anaesthetists named Richard "Harry" Harris appear in the provided texts. One anaesthetist played a crucial role in the Tham Luang cave rescue and was jointly awarded 2019 Australian of the Year as a result, while the other anaesthetist/cave diver has over 30 years of experience, leading teams to record depths and participating in recovery missions and receiving awards for his exceptional contributions to cave diving exploration.
Composer Richard Frank Keith Harris studied composition and orchestration at the University of Edinburgh, co-founded Piano Circus, and commissioned and performed works by Arvo Pärt, Brian Eno, Philip Glass, and Steve Reich. His compositions feature on Argo CDs Loopholes and Landscapes Of The Heart, and he produced successful arrangements of works by Terry Riley and Thomas Ades. His work Hexada was featured in the UK television programme The Score..

Table 18: Examples of irrelevant documents.

