# OpenReview forum: "Lexical Diversity-aware Relevance Assessment for Retrieval-Augmented Generation"
_ICLR.cc/2025/Conference — ICLR 2025 Conference Withdrawn Submission_

### Official Review · Reviewer_CJVL · 2024-10-23

**Soundness:** 3
**Presentation:** 3
**Contribution:** 2
**Rating:** 6
**Confidence:** 4

**Summary:**

This paper proposes Lexical Diversity-aware RAG, which conducts relevance assessment between queries and documents from the perspective of lexical diversity and adopts contrastive decoding to eliminate the adverse effects of irrelevant contents. Experiments on two short-form, two multi-hop and one long-form knowledge-intensive generation/QA tasks show its superior performance.

**Strengths:**

1. The paper provides a fine-grained relevance assessment between queries and retrieved documents by introducing the concepts of lexical diversity, which is both interesting and important. It helps to filter out irrelevant content more accurately, reducing hallucinations and errors in downstream generation.
2. The paper offers valuable insights into document reranking in RAG, emphasizing the importance of delivering high-quality content to improve RAG performance.
3. The experiments show that the proposed method achieves strong performance.

**Weaknesses:**

The paper clearly offers new insights and solutions for the RAG research field. However, this paper has some issues that need to be addressed or clarified:
1. The explanation of the invariant component in Lines 200-203 lacks clarity. For example, "Paris," the capital of France, is often referred to as the "City of Light." If a retrieved document uses "City of Light" without explicitly mentioning "Paris," whether the method proposed is able to handle such issues? Could the author clarify how the proposed method handles cases where "invariant" components have common alternative expressions or propose ways to extend the method to account for such cases?
2. The "supplementary" component serves a valuable purpose, but it is generated directly by the fine-tuned LLM, which inevitably introduces hallucinations. These hallucinations can inject noise into the output, compromising the accuracy of the scoring results. Could the authors also discuss if there are any measures to mitigate this issue or the potential solutions?
3. The contribution of the proposed CRC is limited, as many solutions have explored to utilize the contrastive decoding in RAG to solve similar issues[1][2][3]. Besides, CRC requires parallelly deploy two identical LLMs, this will result in double the deployment costs and computational overhead. Could the authors ptovide more clearly differentiate their CRC approach from existing contrastive decoding methods in RAG? And, the author may also need to clarify the computational efficiency concerns and discuss any potential optimizations or trade-offs.
4. The transferability of the proposed method remains uncertain. The paper leverages data from PopQA and TriviaQA to generate training sets for fine-tuning the model, followed by validation on PopQA, TriviaQA, HotpotQA, 2WikiMultihopQA, and ASQA. While these datasets include a variety of question-answering formats, such as short-form, multi-hop, and long-form, they are all based on wiki-style content. To more convincingly demonstrate the method's effectiveness, the authors should extend their evaluation by testing the trained model on more open and domain-diverse RAG datasets, like FreshQA. Could the author also discuss the potential challenges in applying their method to non-wiki-style datasets and propose specific experiments to show the performance on these datasets?
5. The experiment setup is unclear, and the appendix does not provide additional information: (1) How is factual accuracy calculated? (2) Why is accuracy used as the evaluation metric for the multi-hop dataset, while many prior works, such as SuRe[4], SeaKR[5], and DRAGIN[6], typically use EM/F1 as evaluation metrics? (3) Were all the comparison methods reproduced under the same conditions? (4) What is the source of the retrieval?
6. This paper only compares a few recent RAG methods, and to some extent, the proposed approach is also similar to query expansion/decomposition-based RAG methods. Could the authors propose more comparisons among different RAG methods, such as SeaKR[5], DRAGIN[6], RQ-RAG[7], etc., to help to highlight the differences?
7. Please refer "Questions" section for some minor but noticeable issues.

[1] Zhao et al. Enhancing Contextual Understanding in Large Language Models through Contrastive Decoding

[2] Qiu et al. Entropy-Based Decoding for Retrieval-Augmented Large Language Models

[3] Kim et al. Adaptive Contrastive Decoding in Retrieval-Augmented Generation for Handling Noisy Contexts

[4] Kim et al. SuRe: Summarizing Retrievals using Answer Candidates for Open-domain QA of LLMs

[5] Yao et al. SeaKR: Self-aware Knowledge Retrieval for Adaptive Retrieval Augmented Generation

[6] Su et al. DRAGIN: Dynamic Retrieval Augmented Generation based on the Information Needs of Large Language Models

[7] Chan et al. RQ-RAG: Learning to Refine Queries for Retrieval Augmented Generation

I would like to increase my ratings if the weaknesses can be clearly clarified or addressed.

**Questions:**

1. In Lines 751-752 of the Appendix, the author claims that "manual evaluation indicates GPT-4's predictions align well with human assessments." However, they do not provide any details on the evaluation methods or results to support this claim. Although GPT-4 is highly capable, it still produces hallucinations and incorrect predictions. It is unclear how the author ensured accurate data filtering and quality control to guarantee the correctness of the synthesized data. Could the authors provide specific information about their evaluation process, such as the number of samples evaluated, the criteria used for assessment, and any inter-rater reliability measures if multiple evaluators were involved?
2. Table 3 presents the overall impact of DRA and CRC but lacks a detailed analysis of the effectiveness of individual components within DRA. Specifically, how do the <invariant>, <variant>, and <supplementary> components each contribute to the observed performance improvements? Could the authors conduct an ablation study that isolates the impact of each component on the overall performance?

---

> ### Author Response · Authors · 2024-11-22
> **Reponse to Reviewer CJVL (1/5)**
>
> ### **The question about alternative expressions of "invariant" components**
>
> >***(Weakness 1)*** *The explanation of the invariant component in Lines 200-203 lacks clarity. For example, "Paris," the capital of France, is often referred to as the "City of Light." If a retrieved document uses "City of Light" without explicitly mentioning "Paris," whether the method proposed is able to handle such issues? Could the author clarify how the proposed method handles cases where "invariant" components have common alternative expressions or propose ways to extend the method to account for such cases?*
>
> Thank you for the valuable suggestions regarding invariant components. In cases where an "invariant" component has commonly used alternate expressions, **our DRA module can incorporate these potential aliases within the supplementary components to help capture relevant content more effectively.** Furthermore, a relevant and comprehensive passage often contains multiple components of a query. If an error in analyzing an invariant component impacts relevance assessment, the relevance of other components can still be evaluated, thereby preserving the relevance assessment of the entire query to a certain extent.
>
>
>
> ### **The question about "supplementary" component**
>
> >***(Weakness 2)*** *The "supplementary" component serves a valuable purpose, but it is generated directly by the fine-tuned LLM, which inevitably introduces hallucinations. These hallucinations can inject noise into the output, compromising the accuracy of the scoring results. Could the authors also discuss if there are any measures to mitigate this issue or the potential solutions?*
>
> The "supplementary" component is primarily generated by the fine-tuned LLM and consists of simple descriptions or attributes of other components, without involving complex or in-depth knowledge. This characteristic significantly reduces the likelihood of hallucinations.
>
> Additionally, when assessing the overall relevance of the query, we apply a weighting mechanism to limit the influence of the "supplementary" component, preventing any inaccuracies from disproportionately impacting the evaluation results. In future work, we will explore other techniques for addressing hallucinations, further enhancing the reliability of the DRA module’s generation of "supplementary" components.

---

> ### Author Response · Authors · 2024-11-22
> **Reponse to Reviewer CJVL (2/5)**
>
> ### **The question about CRC module**
>
> >***(Weakness 3)*** *The contribution of the proposed CRC is limited, as many solutions have explored to utilize the contrastive decoding in RAG to solve similar issues[1][2][3]. Besides, CRC requires parallelly deploy two identical LLMs, this will result in double the deployment costs and computational overhead. Could the authors ptovide more clearly differentiate their CRC approach from existing contrastive decoding methods in RAG? And, the author may also need to clarify the computational efficiency concerns and discuss any potential optimizations or trade-offs.*
>
> Thank you for your invaluable suggestions. Below, we provide further clarification on the difference between CRC and other contrastive decoding methods in RAG, as well as the computational efficiency of CRC.
>
> **(1) The difference between CRC and contrastive decoding**
>
> In fact, **other contrastive decoding methods struggle to address the impact of irrelevant content within retrieved documents.** This is because such irrelevant content is often partially related to the query's semantics but does not contribute to answering the query itself. As a result, it can significantly disrupt the reasoning process of LLMs.
>
> **Unlike other contrastive decoding methods, CRC selects retrieved documents that are similar to the query but actually irrelevant, based on relevance assessment, to construct the input for the reference distribution.** This approach ensures that the reference input closely resembles the irrelevant contextual content found within retrieved passages, allowing contrastive decoding to effectively mitigate the impact of irrelevant content in the retrieved documents.
>
> In contrast, methods such as [1][2] utilize contrastive decoding primarily to reduce the impact of the model's internal knowledge, allowing it to focus on retrieved information during output generation. While [3] also aims to eliminate noise, it does so by constructing a completely unrelated noise environment for the query.
>
> In Table 4, we compare our method with a contrastive decoding strategy that utilizes completely unrelated noise text as a reference. The results demonstrate that our approach achieves superior performance, further validating the effectiveness of strongly distracting reference inputs in the CRC module.
>
> [1] Zhao et al. Enhancing Contextual Understanding in Large Language Models through Contrastive Decoding.
>
> [2] Qiu et al. Entropy-Based Decoding for Retrieval-Augmented Large Language Models.
>
> [3] Kim et al. Adaptive Contrastive Decoding in Retrieval-Augmented Generation for Handling Noisy Contexts.
>
> **(2) The Computational efficiency of DRAG**
>
> We can implement CRC by deploying the same LLM and performing parallel inference, thereby reducing additional deployment overhead and inference time. In practical applications, the model used for reference documents in contrastive decoding can be optimized for resource efficiency by utilizing lightweight alternatives. With these deployment strategies, our approach minimizes its impact on system efficiency, ensuring it does not impose a significant negative effect.
>
> Additionally, a significant advantage of our approach is its ability to deliver substantial performance gains without requiring extensive training resources or time. The table below compares the data size and training cost of our method with the training-based RAG approach (Self-RAG), demonstrating the low resource consumption of our method.
>
> Thus, given our method's low training resource demands and the substantial performance gains achieved, the increase in computational load is acceptable.
>
> |||||
> |-|-|-|-|
> ||Number of instances |Model size|Training cost|
> |Self-RAG|145,619|7B|4 Nvidia A100 with 80GB memory|
> |Ours|**6743**|**0.5B**|**1 Nvidia V100 GPU with 24G memory**|
> |||||

---

> ### Author Response · Authors · 2024-11-22
> **Reponse to Reviewer CJVL (3/5)**
>
> ### **The question about transferability to non-wiki-style datasets**
>
> >***(Weakness 4)*** *The transferability of the proposed method remains uncertain. The paper leverages data from PopQA and TriviaQA to generate training sets for fine-tuning the model, followed by validation on PopQA, TriviaQA, HotpotQA, 2WikiMultihopQA, and ASQA. While these datasets include a variety of question-answering formats, such as short-form, multi-hop, and long-form, they are all based on wiki-style content. To more convincingly demonstrate the method's effectiveness, the authors should extend their evaluation by testing the trained model on more open and domain-diverse RAG datasets, like FreshQA. Could the author also discuss the potential challenges in applying their method to non-wiki-style datasets and propose specific experiments to show the performance on these datasets?*
>
> Thank you for your suggestion on improving generalization across different styles. Following your advice, we conducted validation on the FreshQA dataset. The experimental results are shown in the table below:
>
> |||||
> |-|-|-|-|
> ||Llama3-8B|Self-RAG|Ours|
> |ACC(%)|23.0|20.2|**26.2**|
> |||||
>
> The results indicate that our method delivers significant performance improvements on FreshQA, significantly surpassing the baseline methods. This highlights the strong generalization capabilities of our approach across different datasets. Due to time constraints, we have included only the comparison results between our method, Self-RAG, and Llama3-8B with retrieval. More comprehensive experimental comparisons and analyses will be provided in the final version of the paper.
>
> In fact, the non-Wikipedia style of a dataset does not significantly affect the performance of DRAG. This is because DRAG primarily focuses on decoupling the query within the dataset, and as long as the different components of the query can be accurately identified, DRAG can function effectively. However, for highly specialized or domain-specific datasets, such as those used in medical report analysis, domain-specific data may be required to fine-tune the model to understand specialized terminology within queries. Nonetheless, our method requires minimal training data and shows strong potential for effective transfer to such datasets.
>
>
>
> ### **The question about  experiment setup**
>
> >***(Weakness 5)*** *The experiment setup is unclear, and the appendix does not provide additional information: (1) How is factual accuracy calculated? (2) Why is accuracy used as the evaluation metric for the multi-hop dataset, while many prior works, such as SuRe[4], SeaKR[5], and DRAGIN[6], typically use EM/F1 as evaluation metrics? (3) Were all the comparison methods reproduced under the same conditions? (4) What is the source of the retrieval?*
>
> Thank you for your valuable suggestions regarding the explanation of our experimental setup. Following your advice, we will provide further clarification to ensure transparency and precision:
>
> (1) Factual accuracy is computed as follows: if the model's output contains the gold answer, it is considered correct; otherwise, it is deemed incorrect. We have revised the experimental setup in Section 4.1 of the paper to clarify the calculation method for factual accuracy and have highlighted the modifications in red.
>
> (2) Consistent with prior studies [1][2][3], we use factual accuracy as the evaluation metric. Compared to EM/F1 scores, factual accuracy better aligns with the characteristics of open-domain question answering. This is because the output of LLMs is often diverse, and EM/F1 scores may be overly strict when correct answers can have multiple valid expressions, potentially failing to capture the model's true performance.
>
> (3) All methods are evaluated using the same retrieval corpus, retriever, and number of retrieved documents under identical experimental conditions. However, due to challenges in replicating certain training setups, the performance of some methods may differ slightly from the results reported in their original papers.
>
> (4) For PopQA and TriviaQA, we follow prior work by using Wikipedia as the retrieval corpus. For HotpotQA, 2WikiQA, and ASQA, we use the official retrieval documents provided by each dataset to ensure fair comparisons. We have added these information to the experimental details in Appendix C of the paper and highlighted the modifications in red.
>
> We appreciate your feedback and we have incorporated these clarifications in Section 4.1 and Appendix C of the manuscript.

---

> ### Author Response · Authors · 2024-11-22
> **Reponse to Reviewer CJVL (4/5)**
>
> ### **More comparison** **experiment**s
>
> >***(Weakness 6)*** *This paper only compares a few recent RAG methods, and to some extent, the proposed approach is also similar to query expansion/decomposition-based RAG methods. Could the authors propose more comparisons among different RAG methods, such as SeaKR[5], DRAGIN[6], RQ-RAG[7], etc., to help to highlight the differences?*
>
> Thank you for your valuable suggestions. Following your advice, we have supplemented the performance comparison of our method with a few RAG-based methods. The table below provides a performance comparison of our method with RQ-RAG[1], QD-RAG [2], and CtrlA [3] on the PopQA dataset. While we conducted experiments on SeaKR [4] and DRAGIN [5], the experiments are not yet fully completed due to differences in experimental settings and time constraints. We will include more comprehensive comparative experiments in the final version of the paper.
>
> |||||||
> |-|-|-|-|-|-|
> ||RQ-RAG |QD-RAG|CtrlA|Ours-Llama2|Ours-LLama3|
> |acc|57.1|29.4|61.8|67.0|68.5|
> |||||||
>
> The experimental results demonstrate that our method achieves superior performance, further underscoring the effectiveness of our proposed approach for query decoupling and fine-grained relevance assessment.
>
> **Distinction Between Our Method and Other Query Expansion/Decomposition-Based RAG:**
>
> The objectives and processes of our DRA module and Other Query Expansion/Decomposition-Based RAG are fundamentally distinct, specifically:
>
> (1) The Query Expansion/Decomposition-Based RAG methods does not account for lexical diversity, which limits its ability to address the challenges of inaccurate relevance assessment caused by diverse lexical expressions.
>
> (2) The DRA module decouples a query into multiple distinct components (at the word or phrase level) based on lexical diversity, making it applicable to both single-hop and multi-hop questions. In contrast, Query Decomposition-Based RAG methods (such as the query decomposition operation in RQ-RAG[1]) primarily focus on splitting multi-hop questions into multiple sub-queries for step-by-step reasoning.
>
> (3) The DRA module can be integrated with Query Expansion/Decomposition-Based RAG methods, allowing for fine-grained decoupling and relevance evaluation of different components within the expanded queries or decomposed sub-queries. This combination enhances the overall performance by leveraging the strengths of both approaches.
>
> [1] Chan et al. RQ-RAG: Learning to Refine Queries for Retrieval Augmented Generatio.
>
> [2] Khattab O, Santhanam K, Li X L, et al. Demonstrate-search-predict: Composing retrieval and language models for knowledge-intensive nlp.
>
> [3] Liu H, Zhang H, Guo Z, et al. CtrlA: Adaptive Retrieval-Augmented Generation via Probe-Guided Control.
>
> [4] Yao et al. SeaKR: Self-aware Knowledge Retrieval for Adaptive Retrieval Augmented Generation.
>
> [5] Su et al. DRAGIN: Dynamic Retrieval Augmented Generation based on the Information Needs of Large Language Models.

---

> ### Author Response · Authors · 2024-11-22
> **Reponse to Reviewer CJVL (5/5)**
>
> ### **Human validation of synthesized data**
>
> >***(Question 1)*** *In Lines 751-752 of the Appendix, the author claims that "manual evaluation indicates GPT-4's predictions align well with human assessments." However, they do not provide any details on the evaluation methods or results to support this claim. Although GPT-4 is highly capable, it still produces hallucinations and incorrect predictions. It is unclear how the author ensured accurate data filtering and quality control to guarantee the correctness of the synthesized data. Could the authors provide specific information about their evaluation process, such as the number of samples evaluated, the criteria used for assessment, and any inter-rater reliability measures if multiple evaluators were involved?*
>
> Thank you for your suggestions regarding the accuracy of generated data. **In fact, our method does not rely on highly precise and strictly accurate generated data, eliminating the need for extensive manual annotation and validation. This is also one of the key advantages of our approach.** This is because the purpose of using generated data for training is to equip our first module (DRA) with the ability to perform query decoupling and relevance analysis based on lexical diversity. As long as the generated data reasonably reflects the designed analysis and processing framework, minor inaccuracies in the data will not significantly impact the performance of our method.
>
> During experiments, we observed some minor inaccuracies in the data, such as certain explanatory nouns in the question being categorized as "supplementary" components instead of "invariant" components as per the design. However, despite these issues, our method still achieved significant performance improvements. This strongly demonstrates that our approach is robust and does not depend on strictly annotated data.
>
> Additionally, following your suggestions, we conducted further validation to ensure that the generated data reasonably reflects our design regarding lexical diversity. We sampled a subset consisting of 100 query decoupling data points and 500 relevance assessment data points for cross-validation by human evaluators. The evaluators included three data annotation experts and two of our co-authors. They were tasked with evaluating the data quality based on both the correctness of the data and the logical consistency of the GPT reasoning process. Each evaluator's assessment was cross-validated by two other evaluators. The results showed that the generated data largely meets the reasoning requirements of DRAG. We have included 10 examples each of query decoupling data and relevance assessment data in the appendix. From these samples, it is evident that our generated data is generally accurate and reasonable, effectively capturing the logical reasoning processes involved in query decoupling and relevance assessment.
>
> In future work, we will explore more robust data generation approaches and implement stricter methods for verifying data accuracy to further enhance the reliability of our training data.
>
>
>
> ### **Effectiveness of individual components within DRA.**
>
> > ***(Question 2)*** *Table 3 presents the overall impact of DRA and CRC but lacks a detailed analysis of the effectiveness of individual components within DRA. Specifically, how do the <invariant>, <variant>, and <supplementary> components each contribute to the observed performance improvements? Could the authors conduct an ablation study that isolates the impact of each component on the overall performance?*
>
> Thank you for your insightful suggestions regarding the refinement of ablation experiments. Following your advice, we conducted a more detailed analysis of the contributions of each category of components within DRA. The experimental results are presented in the table below.
>
> ||||||||
> |-|-|-|-|-|-|-|
> ||LLama3 with retrieval|w/o DRA|Invariant only|Variant only|Supplementary only |ours|
> |PopQA|62.7|65.0|67.9|66.2|65.6|**68.5**|
> ||||||||
>
> The results demonstrate that each category contributes to the overall performance of our method. Notably, the performance contributions of the invariant, variant, and supplementary components decrease sequentially. This observation aligns with our hypothesis that the importance of these three categories for relevance assessment diminishes in the same order. We have included this detailed ablation experiment and analysis in the Appendix C of the paper.

---

### Official Review · Reviewer_eYsj · 2024-10-28

**Soundness:** 2
**Presentation:** 3
**Contribution:** 3
**Rating:** 5
**Confidence:** 4

**Summary:**

The paper introduces a Lexical DRAG model, aiming to enhance LLMs by addressing the limitations of previous RAG methods. DRAG incorporates a DRA module and a CFC module to refine the relevance assessment between queries and retrieved documents. The method significantly improves performance by ensuring a more granular and accurate relevance evaluation, leading to a 12.4% accuracy boost on the HotpotQA benchmark.

**Strengths:**

- The focus on improving the diversity of the retrieved documents in RAG systems is promising and practically beneficial, addressing a critical problem in the field.
- The method’s effectiveness is clearly demonstrated through substantial experiments, showing notable improvements in accuracy across several benchmarks.
- The paper is well-written, and the idea is easy to understand and follow.

**Weaknesses:**

- While the method is innovative, its technical depth appears limited, relying significantly on GPT-4 data augmentation. The application of contrastive relevance calibration could be viewed as a nuanced extension of contrastive decoding, lacking substantial novelty.
- The validation for the correctness of generated data is missing. Effective validation would ideally include rigorous human annotation to ensure the quality and factual accuracy of the augmented data.
- The need for an additional module to decompose the query and multiple retrieval processes introduces considerable computational costs, which might limit the method’s applicability in real applications. This aspect, along with its impact on system efficiency, should be discussed.
- The model’s applicability to complex, multi-hop question answering scenarios (e.g., identifying sequential factual relationships) seems limited. When a sub-query is dependent on the other one, it is hard to retrieve relevant documents in a single retrieval step.

**Questions:**

Please consider to reply to my concerns in weaknesses.

---

> ### Author Response · Authors · 2024-11-22
> **Reponse to Reviewer eYsj (1/2)**
>
> ### **The question about innovation**
>
> >***(Weakness 1)*** *While the method is innovative, its technical depth appears limited, relying significantly on GPT-4 data augmentation. The application of contrastive relevance calibration could be viewed as a nuanced extension of contrastive decoding, lacking substantial novelty.*
>
> Thank you for your suggestions. **In fact, GPT-4 data augmentation is not the focus of our method. The core challenge and technical depth of our approach lie in designing and implementing evaluation strategies to address the issue of inaccurate relevance assessments caused by lexical diversity, a common problem in RAG systems.** To tackle this, we analyzed typical query structures in open-domain question answering and categorized query components based on lexical diversity. We then developed a hierarchical and fine-grained evaluation strategy to accurately assess the relevance of retrieved documents by analyzing the different components of the query.
>
> Additionally, our CRC module is not a simple application of contrastive decoding. Unlike conventional contrastive decoding methods, CRC constructs reference inputs by selecting documents that exhibit a certain degree of embedding similarity to the query but are, in reality, irrelevant. This ensures that the reference input closely mirrors the type of irrelevant contextual content typically found within genuinely relevant passages. This process effectively minimizes the impact of unrelated information on LLM inference.
>
> In Table 4 of the paper, we provide a comparison with other contrastive decoding strategies. The results demonstrate that constructing strongly distracting reference documents as we propose significantly enhances performance.
>
> ### **The validation for the correctness of generated data**
>
> >***(Weakness 2)*** *The validation for the correctness of generated data is missing. Effective validation would ideally include rigorous human annotation to ensure the quality and factual accuracy of the augmented data.*
>
> Thank you for your suggestions regarding the accuracy of generated data. **In fact, our method does not rely on highly precise and strictly accurate generated data, eliminating the need for extensive manual annotation and validation. This is also one of the key advantages of our approach.** This is because the purpose of using generated data for training is to equip our first module (DRA) with the ability to perform query decoupling and relevance analysis based on lexical diversity. As long as the generated data reasonably reflects the designed analysis and processing framework, minor inaccuracies in the data will not significantly impact the performance of our method.
>
> During experiments, we observed some minor inaccuracies in the data, such as certain explanatory nouns in the question being categorized as "supplementary" components instead of "invariant" components as per the design. However, despite these issues, our method still achieved significant performance improvements. This strongly demonstrates that our approach is robust and does not depend on strictly annotated data.
>
> Additionally, following your suggestions, we conducted further validation to ensure that the generated data reasonably reflects our design regarding lexical diversity. A subset of the data, consisting of 100 query decoupling data points and 500 relevance assessment data points, was sampled for cross-validation by human evaluators. The evaluation team comprised three data annotation experts and two of our co-authors. They assessed the data quality based on its correctness and the logical consistency of the GPT-generated reasoning process. Each evaluation was cross-validated by two other evaluators. The results confirmed that the generated data largely satisfies the reasoning requirements of DRAG. We have included 10 sample cases in the revised Appendix D. These instances indicate that the generated data is largely accurate and reasonable, effectively reflecting the logical reasoning processes involved in query decoupling and relevance assessment.
>
> In fact, using GPT-4 to construct data is widely regarded as a highly accurate data augmentation method[1][2][3]. Some studies have shown that GPT-4's annotation capabilities even surpass human performance, ensuring a high level of accuracy in generated data [4]. In future work, we plan to implement stricter methods for verifying data accuracy to further enhance the reliability of our training data.
>
> [1] Asai A, Wu Z, Wang Y, et al. Self-rag: Learning to retrieve, generate, and critique through self-reflection. ICLR 2024.
>
> [2] Liu Y, Iter D, Xu Y, et al. G-eval: Nlg evaluation using gpt-4 with better human alignment.
>
> [3] Pei X, Li Y, Xu C. Gpt self-supervision for a better data annotator.
>
> [4] Pan A, Chan J S, Zou A, et al. Do the rewards justify the means? measuring trade-offs between rewards and ethical behavior in the machiavelli benchmark. ICML 2023.

---

> ### Author Response · Authors · 2024-11-22
> **Reponse to Reviewer eYsj (2/2)**
>
> ### **The question about computation**
>
> >***(Weakness 3)*** *The need for an additional module to decompose the query and multiple retrieval processes introduces considerable computational costs, which might limit the method’s applicability in real applications. This aspect, along with its impact on system efficiency, should be discussed.*
>
> Thank you for the insightful suggestions on computation. **In fact, our module primarily focuses on query decomposition and relevance assessment, which does not increase the number of retrieval operations.** Furthermore, as our method relies on additional modules and inference processes, the additional computational overhead can be mitigated by employing parallel processing to reduce inference time. Consequently, our approach does not have a significant negative impact on system efficiency.
>
> Moreover, a key advantage of our approach is that it requires minimal training resources and time to achieve substantial performance improvements. The table below compares the data size and training cost of our method with the training-based RAG approach (Self-RAG), demonstrating the low resource consumption of our method.
>
> |||||
> |-|-|-|-|
> ||Number of instances |Model size|Training cost|
> |Self-RAG|145,619|7B|4 Nvidia A100 with 80GB memory|
> |Ours|**6743**|**0.5B**|**1 Nvidia V100 GPU with 24G memory**|
> |||||
>
>
>
> ### **The applicability to complex, multi-hop scenarios**
>
> >***(Weakness 4)*** *The model’s applicability to complex, multi-hop question answering scenarios (e.g., identifying sequential factual relationships) seems limited. When a sub-query is dependent on the other one, it is hard to retrieve relevant documents in a single retrieval step.*
>
> Thank you for your suggestions. **In fact, we conducted experiments on multi-hop questions in our study. As shown in Table 1, our method achieved performance improvements of 12.4% and 11.5% on the multi-hop datasets HotpotQA and 2WikiMultiHopQA, respectively.** This provides strong evidence that our approach is well-suited for handling complex multi-hop problems.
>
> Actually, for multi-hop questions, DRAG can address the challenges through two approaches:
>
> (1) Direct Decoupling: DRAG can directly decouple the original complex query to perform fine-grained evaluations of the relevance between retrieved documents and the various components involved in the multi-hop reasoning process. This enables the retrieval of most of the required information for multi-hop questions in a single step.
>
> (2) Combination with Query Decomposition: DRAG can be combined with query decomposition methods to evaluate the retrieval relevance of each sub-query step by step. This approach is particularly effective in scenarios where one sub-query depends on the results of another.

---

> ### Author Response · Authors · 2024-12-01
>
> We appreciate your feedback! More discussions are welcomed if you possess further questions and suggestions.

---

### Official Review · Reviewer_pzHU · 2024-11-04

**Soundness:** 3
**Presentation:** 3
**Contribution:** 3
**Rating:** 5
**Confidence:** 4

**Summary:**

This paper introduces a Lexical Diversity-aware RAG (DRAG) model, comprising a Diversity-sensitive Relevance Analyzer (DRA) and a Contrastive Relevance Calibration Module (CRC).

DRAG: A small model trained to decouple queries and classify the relevance of retrieved documents.

CRC: Calibrate the model’s decoding process by contrasting it with outputs under noise information.

**Strengths:**

This paper explores word-level decomposition for Retrieval-Augmented Generation (RAG) and introduces a method to train Large Language Models (LLMs) to effectively ignore noise in the context.

**Weaknesses:**

This paper proposed a good solution for word-level decomposition. However, I have two questions about the evaluation process.

1. The first stage just involves using a trained LLM to process the query, which improves the performance of multihop-query. This module works great.  However, how about this module compare to normal query decompose? The processes between these two methods are highly similar.

2. For the second stage, CRC, how about this process compared to a normal LLM + Ranker? They share the same function. The improvement gain of CRC in Table 3 is marginal.

**Questions:**

The same with the weakness section.

---

> ### Author Response · Authors · 2024-11-22
> **Reponse to Reviewer pzHU (1/1)**
>
> Our sincere thanks for the in-depth review of our paper. We have carefully tracked and resolved each issue put forward by them.
> ### **The comparison of  DRA and query decompose methods**
>
> > ***(Weakness 1)*** *The first stage just involves using a trained LLM to process the query, which improves the performance of multihop-query. This module works great. However, how about this module compare to normal query decompose? The processes between these two methods are highly similar.*
>
> We appreciate your attention and feedback. **The objectives and processes of our first stage (the DRA module) and the Query Decomposition RAG method (QD-RAG) are fundamentally distinct:**
>
> (1) The DRA module decouples a query into multiple distinct components (at the word or phrase level) based on lexical diversity, making it suitable for both single-hop and multi-hop questions. In contrast, QD-RAG primarily focuses on splitting multi-hop questions into multiple sub-queries.
>
> (2) The QD-RAG method does not account for lexical diversity, which limits its ability to address the challenges of inaccurate relevance assessment caused by diverse lexical expressions.
>
> (3) The DRA module can be combined with query decomposition methods, enabling fine-grained decoupling and relevance evaluation of different components within each decomposed sub-query, thus enhancing overall performance.
>
>
>
> ### **The comparison of  CRC and  normal LLM + Ranker**
>
> > ***(Weakness 2)*** *For the second stage, CRC, how about this process compared to a normal LLM + Ranker? They share the same function. The improvement gain of CRC in Table 3 is marginal.*
>
> Thank you for your suggestions. The functionality of the CRC module differs from that of LLM+Ranker.  **While LLM+Ranker is used to re-rank retrieved documents, CRC mitigates the influence of irrelevant noise during model inference by constructing reference documents.**
>
> In addition, our overall approach is distinctly different from LLM+Ranker. Our method uniquely incorporates the impact of lexical diversity, enabling finer-grained relevance assessments. In contrast, LLM+Ranker relies solely on the overall query semantics, overlooking the influence of lexical diversity. In Appendix C, we present a performance comparison between our approach and methods combining LLM with various rankers.
>
> **Additionally, although CRC alone provides limited performance improvement as shown in Table 3, combining it with DRA leads to further performance gains. On the PopQA dataset, CRC+DRA achieves a significant 4.5% increase in accuracy compared to DRA alone.** This improvement is due to CRC's ability to construct reference input by selecting retrieved documents that share some similarity with the query but are actually irrelevant. This effectively simulates the presence of strong distracting irrelevant information within the retrieved documents, thereby mitigating its impact through contrastive decoding.

---

> > ### Comment · Reviewer_pzHU · 2024-11-25
> >
> > Thanks for your response. I prefer to maintain my score.

---

> > > ### Author Response · Authors · 2024-12-01
> > >
> > > Thanks for your feedback! We're grateful for your constructive comments. We'd be glad to have more discussions in case you have additional questions and suggestions.

---

### Official Review · Reviewer_af8N · 2024-11-04

**Soundness:** 3
**Presentation:** 3
**Contribution:** 2
**Rating:** 5
**Confidence:** 3

**Summary:**

This paper proposes a Lexical Diversity-aware RAG framework. In DRA module, the author tailors query decoupling task and corresponding relevance assessment to eliminate the negative impact caused by lexical diversity, applying a granular retrieval and reranking process. Then, the contrastive decoding method is introduced in CRC module for subsequent answer generation, utilizing the result from DRA module. The proposed framework is evaluated by numerical experiments on three tasks across multiple datasets.

**Strengths:**

1.	It raises the question about how lexical diversity can influence the retrieval stage, and proposes a comprehensive framework to handle it.
2.	I believe the proposal of query decoupling part is good, it covers three attributes from a reasonable perspective.
3.	The CRC utilizes the assessment from DRA, which is a very natural design.

**Weaknesses:**

1.	The paper mainly follows the standard approach of training models with generated data and then performing inference. The contrastive decoding method is just integrated within the framework. From this perspective, it has limited novelty.
2.	The author only provides experiments to demonstrate that the proposed method outperforms others, but there is a lack of detailed experiments and analysis showing how the identified issue of lexical diversity is addressed by the proposed framework.

**Questions:**

1.	Is it necessary to train the model component of DRA in different scenarios? If it is intended to be general, how well does this module generalize across different datasets?
2.	In the appendix C, comparing other retrieval models, why weren’t more commonly used retrieval models selected? Based on the experiments presented, it seems hard to see a significant improvement in retrieval performance using the DRA component.

---

> ### Author Response · Authors · 2024-11-22
> **Reponse to Reviewer af8N (1/3)**
>
> Thank you so much for your insightful questions and suggestions. We have carefully followed and cleared up every issue raised by them.
>
> ### **The innovation of Our method**
>
> > ***(Weakness 1)*** *The paper mainly follows the standard approach of training models with generated data and then performing inference. The contrastive decoding method is just integrated within the framework. From this perspective, it has limited novelty.*
>
> We appreciate your insightful suggestions. In fact, the standard approach of training models with generated data and the use of contrastive decoding are not the core innovations of our work.
>
> The innovation of DRAG lies in two key aspects: **The first module, DRA, introduces lexical diversity for the first time** to enable fine-grained relevance assessment of retrieved documents. **The second module, CRC, constructs strongly distracting reference inputs**, enabling contrastive decoding to effectively mitigate the influence of irrelevant content—**an outcome that other contrastive decoding methods are unable to achieve.** Specifically:
>
> (1) The innovation of DRA:
>
> The innovation of the DRA module lies in introducing lexical diversity into retrieval relevance assessment for the first time, rather than the standard training approach with generated data. Specifically, by decoupling the query into distinct components and defining differentiated evaluation criteria, DRA enables fine-grained and precise assessments of retrieved documents, significantly improving the ability of RAG systems to retrieve truly relevant information.
>
> (2) The innovation of CRC:
>
> Unlike other contrastive decoding methods, CRC leverages relevance assessment results to select documents with a certain degree of embedding similarity to the query but are actually irrelevant, using them to construct input for the reference distribution. This approach ensures that the reference input closely mirrors the irrelevant contextual content within retrieval passages, thereby enabling contrastive decoding to effectively mitigate the impact of irrelevant content in the retrieved documents. In Table 4 of the paper, we provide a comparison with other contrastive decoding strategies. The results demonstrate that constructing strongly distracting reference documents as we propose significantly enhances performance.
>
> We will further refine and clarify the corresponding descriptions in the paper to better highlight our core innovations.

---

> ### Author Response · Authors · 2024-11-22
> **Reponse to Reviewer af8N (2/3)**
>
> ### **The experiment about lexical diversity**
>
> > ***(Weakness 2)*** *The author only provides experiments to demonstrate that the proposed method outperforms others, but there is a lack of detailed experiments and analysis showing how the identified issue of lexical diversity is addressed by the proposed framework.*
>
> Thank you for your valuable suggestions. Acctually, in our experiments, **a key indicator of addressing the issue of lexical diversity is that**, after incorporating fine-grained relevance assessment based on lexical diversity, **the RAG system exhibits improved effectiveness in retrieving genuinely relevant documents and generating accurate answers from their content**.
>
> As illustrated in Table 3 of the paper, **the integration of lexical diversity** (the row labeled "w/o CRC") leads to significant performance improvements, **including an 11.3% increase in accuracy on HotpotQA**. This provides compelling evidence that the issue of lexical diversity has been effectively mitigated.
>
> Furthermore, following your advice, we evaluated the contribution of different components within DRA to retrieval performance and provided a detailed case study to demonstrate how our method effectively addresses the challenges posed by lexical diversity.
>
> (1) First, **we validated on the PopQA dataset the performance contributions of the different types of components extracted from the query to demonstrate that these components help retrieve information with varying lexical diversity**, as shown in the table below. It can be observed that the performance associated with each type of query component significantly exceeds that of the approach without considering lexical diversity (w/o DRA). This demonstrates that our method's consideration of lexical diversity effectively aids in retrieving genuinely relevant documents and generating accurate answers.
>
> ||||||||
> |-|-|-|-|-|-|-|
> ||LLama3-RAG|w/o DRA|Invariant only|Variant only|Supplementary only |ours|
> |PopQA|62.7|65.0|67.9|66.2|65.6|68.5|
> ||||||||
>
> (2) Additionally, we present a case study illustrating how our method addresses the challenge of lexical diversity. In the example, a query and two relevant document passages are analyzed. The baseline retrieval fails to maintain the lexical integrity of "Arcangelo Ghisleri," instead being misled by repeated occurrences of "Arcangelo" in unrelated contexts, resulting in irrelevant documents. Our method, by analyzing lexical diversity and applying refined evaluation criteria, identifies the passage that exactly matches "Arcangelo Ghisleri" and implicitly references "occupation" through the term "journalist." This ensures the retrieval of the correct document, enabling accurate model outputs.
>
> |||
> |-|-|
> |**Question**|What is Arcangelo Ghisleri's occupation?|
> |The top 1 retrieval document of **baseline**|S. Michele Arcangelo, archangel in Jewish, Christian, and Islamic teachings ; Andrea di Cione Arcangelo (1308–1368), Italian painter, sculptor, and architect active in Florence ; Antonio di Arcangelo, Italian painter, active in Florence in a Renaissance style, between 1520 and 1538 ; Arcangelo Califano (1730–1750), baroque composer and cellist ; Arcangelo Placenza da Calatafimi, (1390–1460) venerated Italian Franciscan friar and preacher ... |
> |**Answer of Baseline**|Arcangelo Ghisleri is a geographer who created numerous maps of Africa.|
> |The top 1 retrieval document of **Ours**|Ghislieri ; Michele Ghislieri (1504–1572), also known as Pope Pius V. ; Ghislieri College ; Ghislieri Choir and Consort, Giulio Prandi Ghisleri ; **Arcangelo Ghisleri (1855–1938), an Italian journalist.** The Ghislieri, or less commonly Ghisleri, were an ancient Bolognese aristocratic family.|
> |**Answer of ours**|Arcangelo Ghisleri (1855–1938) is an Italian journalist.|
> |||
>
> We have included these experiments and analyses in Appendix E to provide a more comprehensive discussion of lexical diversity.

---

> ### Author Response · Authors · 2024-11-22
> **Reponse to Reviewer af8N (3/3)**
>
> ### **The experiment about generalization**
>
> >***(Question 1)*** *Is it necessary to train the model component of DRA in different scenarios? If it is intended to be general, how well does this module generalize across different datasets?*
>
> Thanks for sharing the suggestions and insights about module generalization. **For common open-domain QA tasks, there is no need to train the model component of DRA separately for different scenarios.** The experimental results of our method on datasets with various task types, as presented in Table 1 of the paper, provide strong evidence supporting its effectiveness. Specifically, our method achieves performance improvements of 12.4% and 11.5% on the multi-hop datasets HotpotQA and 2WikiMultiHopQA, respectively, while delivering state-of-the-art performance on the long-text generation dataset ASQA.
>
> This is because the characteristics of the three components involved in our method are generally consistent across scenarios. Specifically, entities or specific nouns are typically invariant components, descriptions of non-entities are often invariant components, and supplementary components usually provide additional information about the first two. As a result, datasets like PopQA and TriviaQA, which we used, are broadly applicable to most open-domain QA tasks.  For highly specialized or niche QA tasks, such as medical report analysis, domain-specific data may be required to train the model for accurate query decomposition. However, our method requires minimal training data and has strong potential for effective transfer to these downstream domains.
>
> Additionally, we evaluated the performance on FreshQA [1], a different style dataset, as shown in the table below. The experimental results demonstrates the strong generalization capability of our approach across other datasets.
>
> |||||
> |-|-|-|-|
> ||Llama3-8B with retrieval|Self-RAG|Ours|
> |ACC(%)|23.0|20.2|**26.2**|
> |||||
>
> In future research, we will follow your suggestion to further explore its application in additional scenarios, such as multi-turn dialogue and medical report analysis.
>
> [1] Vu T, Iyyer M, Wang X, et al. Freshllms: Refreshing large language models with search engine augmentation.
>
> ### **The question about appendix C**
>
> > ***(Question 2)*** *In the appendix C, comparing other retrieval models, why weren’t more commonly used retrieval models selected? Based on the experiments presented, it seems hard to see a significant improvement in retrieval performance using the DRA component.*
>
> Thank you for your valuable suggestions. In fact, as outlined in Appendix C, **the objective of our experiments was to compare the performance of DRAG with traditional embedding-based re-ranking methods, rather than with retrieval models.**
>
> Specifically, in the experiment of appendix C, we first used standard retrievers (either Contriever or official BM25 retrieval) to obtain a preliminary set of potentially relevant documents. We then applied typical re-ranking models (monoBERT and monoT5) to re-rank these initial results, selecting the top *r* most relevant inputs to pass to the LLM for output generation. In Table 6 of the paper, the first row represents the results when the initial retrieved documents are directly input to the LLM without re-ranking, while monoBERT and monoT5 are two representative re-ranking models. The experimental results demonstrate that our approach achieves superior performance.

---

### Author Response · Authors · 2024-11-22
**Response to all Reviewers**

We thank all the reviewers for their thoughtful and valuable feedback! We have conducted additional experiments and revised the paper based on the reviews. We highlighted all changes in red. We also included more results and analysis to make the paper more comprehensive. For ease of reading, we have appended the Appendix, which was previously included in the Supplementary Material, to the end of the main text.

Below is a summary of the main changes. Please let us know if you have further questions.



||||
|-|-|-|
|Change |Section|Related Reviewers|
|Analysis of lexical diversity|Appendix E|Reviewer af8N|
|More analysis of generalization|Appendix C|Reviewer af8N, Reviewer CJVL|
|Analysis of efficiency|Appendix C|Reviewer eYsj, Reviewer CJVL|
|Analysis of training data|Appendix D|Reviewer eYsj, Reviewer CJVL|
|Add more details of experimental setup|Section 4.1 and Appendix C|Reviewer CJVL|
|Analysis of individual components within DRA|Appendix C|Reviewer CJVL|
||||

---

### Note · Authors · 2024-12-15

I have read and agree with the venue's withdrawal policy on behalf of myself and my co-authors.